# How to Verify Any (Reasonable) Distribution Property: Computationally Sound Argument Systems for Distributions

**Tal Herman**
Weizmann Institute of Science
Rehovot, Israel
{tal.herman}@weizmann.ac.il

**Guy N. Rothblum**
Apple
{rothblum}@alum.mit.edu

## Abstract

As statistical analyses become increasingly central, there is a growing need to ensure their results are correct. Approximate correctness can be verified by replicating the entire analysis, but can we verify without replication? We focus on distribution testing problems: given samples from an unknown distribution, the goal is verifying that the distribution is close to having a claimed property. Our main contribution is an interactive protocol between a verifier and an untrusted prover who both have sampling access to the unknown distribution. Our protocol can be used to verify a very rich class of properties: the only requirement is that, given a full and explicit description of a distribution, it should be possible to approximate its distance from the property in polynomial time. For any such property, if the distribution is at statistical distance $\varepsilon$ from having the property, then the verifier rejects with high probability. This soundness property holds against any polynomial-time strategy that a cheating prover might follow, assuming the existence of collision-resistant hash functions (a standard assumption in cryptography). For distributions over a domain of size $N$, the protocol consists of $4$ messages and the communication complexity and verifier runtime are roughly $\widetilde{O}\left(\sqrt{N}/\varepsilon^2\right)$. The verifier's sample complexity is $\widetilde{O}\left(\sqrt{N}/\varepsilon^2\right)$, and this is optimal up to $\mathsf{polylog}(N)$ factors (for any protocol, regardless of its communication complexity). Even for simple properties, approximately deciding whether an unknown distribution has the property can require quasi-linear sample complexity and running time. For any such property, our protocol provides a quadratic speedup over replicating the analysis.

## 1 Introduction

Statistical analyses on valuable datasets routinely guide high-stake decisions in domains as diverse as public policy, medicine and finance, and their importance is growing rapidly. As the stakes for data science computations grow, it is increasingly important to obtain assurances that the results of the analyses are correct. An emerging line of work takes a novel approach to tackling these questions: using *proof-systems* from the theoretical computer science literature to supply efficiently-verifiable proofs for the correctness of data science computations.

We focus on a setting where we have i.i.d. sampling access to an unknown distribution, and we want to learn about the properties of the distribution. We can learn by collecting samples from the distribution and performing a sophisticated analysis, but this is expensive, both in terms of sample complexity and the computational complexity of performing the analysis on the collected dataset. Suppose that an untrusted data analysis firm claims to have painstakingly assembled a large collection of samples drawn from the distribution, to have run the analysis, and arrived at some conclusions (for a price, of course). Can the firm provide a proof that would let us verify their claims about the distribution's properties? If we don't have *any* access to the distribution, then verification is impossible, but can we verify using fewer samples and computational resources than we would needed to perform the analysis ourselves (i.e. without replicating the analysis)?

In this work, we focus on interactive and probabilistic *computationally sound* proof systems. The proof is a protocol between the verifier and an untrusted prover, who both have sampling access to the unknown distribution. The prover's claim is that the distribution has (or is close to having) a property, as in the distribution testing literature. If the prover's claim is correct and it follows the protocol, then the verifier accepts with high probability. If the claim is *far* from correct, i.e. the distribution is far from the property, then no polynomial-time cheating prover can make the verifier accept with non-negligible probability. This latter *soundness* property is proved under (standard) cryptographic assumptions: in particular, assuming the existence of hash functions where it is computationally hard to find collisions (collision-resistant hash function families, see Definition A.9). Our focus on distribution properties draws inspiration from the property testing literature Goldreich et al. (1998); Rubinfeld & Sudan (1996) and builds on a study of *unconditionally sound* proof systems for distribution properties initiated by Chiesa and Gur Chiesa & Gur (2018) and developed in recent works of Herman and Rothblum Herman & Rothblum (2022; 2023; 2024). Following the cryptographic literature, we refer to computationally sound protocols as (interactive) *argument systems*, whereas unconditionally sound protocols are referred to as (interactive) *proofs*.

We aim to construct argument systems where the verifier is as efficient as possible: the primary resources we consider (and aim to minimize) are the verifier's sample complexity and running time, as well as the protocol's communication and round complexities. In particular, the verifier's sample complexity should be significantly lower than the sample complexity needed to decide whether the distribution is (close to being) in the property in the standalone setting (i.e. without a prover). On the (honest) prover's side, we also want the time and sample complexities needed for generating the proof to be as small as possible. The proof system should be *doubly-efficient* Goldwasser et al. (2015); Herman & Rothblum (2023): generating the proof should be possible in polynomial time and sample complexities.

## 1.1 Our Results: Argument Systems for General Distribution Properties

We show that essentially any reasonable property (see below) has an efficiently verifiable argument system. Taking $N$ to be a bound on the size of the domain, the communication and the verifier's sample complexity are $\widetilde{O}(\sqrt{N})$. For many natural properties, the sample complexity needed to decide the problem in the standalone setting (without an untrusted prover) is quasi-linear in $N$, so our protocol provides a quadratic speedup over replicating the analysis.

We proceed to detail this result: a *distribution property* is a set of distributions (similarly to the way a *language* is a set of strings), parameterized by the size $N$ of the domain. We measure the distance of a distribution $D$ from a property $\Pi$ by $D$'s total variation distance to the closest distribution in $\Pi$. We study families of properties characterized by the computational complexity of deciding, given the full specification of a distribution, whether it is $\delta_c$-close to the property (the YES case), or $\delta_f$-far from the property (the NO case). A Turing Machine for this problem gets as its input the parameters $\delta_c, \delta_f$ and a list $((i, D[i]))_{i \in [N]}$ specifying the probability of each element in the support (probabilities are discretized to $\text{poly}(1/N)$ precision).[1] We emphasize that this is a purely computational problem: the distribution is fully specified, there is no need to draw samples. We say that a distribution property can be $\rho(N)$-approximately decided (or "approximated") in polynomial time if there exists a poly-time Turing machine that decides the aforementioned decision problem for every $\delta_c, \delta_f$ s.t. $(\delta_f - \delta_c) \geq \rho(N)$. This is an incredibly rich family of distribution properties: it captures essentially any property that can be reasoned about in $\text{poly}(N)$-time (see Appendix A.1 for a formal definition).[2]

**Theorem 1.1 (Main result: computationally sound arguments for all efficient properties)**
*Assume the existence of a collision-resistant hash family (Definition A.9) and let $\kappa = \kappa(N)$ be a*

---

[1] One can consider different natural representations. Many of these are equivalent under polynomial-time reductions. They are thus interchangable for polynomial-time Turing machines, and we fix the above representation.

[2] For examples, distribution properties that are *not efficiently decidable* include the property of all distributions supported only on a noncomputable set of integers. Another example is the distribution property defined through a family of graphs parameterized by $N$, with $[N]$ vertices: consider the distribution property of being supported over a $N/2$-clique of the respective graph for every $N$. Since this problem is NP-complete, it is efficiently decidable only if P=NP.

*cryptographic security parameter. For every approximation parameter $\rho = \rho(N) \in ((1/\sqrt{N}), 1),$[3] and every property that can be $\rho(N)$-approximately decided in $\mathsf{poly}(N)$ time, there is a computationally-sound argument system as follows. The prover and the verifier get as input an integer $N$ and proximity parameters $\epsilon_c, \epsilon_f \in [0, 1]$ s.t. $\epsilon_f - \epsilon_c \geq \rho$, and sampling access to a distribution $D$ over $[N]$, where*

- *Completeness: if $D$ is $\epsilon_c$-close to the property and the prover follows the protocol, then the verifier accepts with all but negligible probability (smaller than any inverse polynomial).*

- *Soundness: if $D$ is $\epsilon_f$-far from the property, then, for every $\mathsf{poly}(N, \kappa)$-time non-uniform cheating prover $\widetilde{P}$, the probability that the verifier accepts in its interaction with $\widetilde{P}$ is negligible.*

- *Efficient verification: the communication complexity, the verifier's sample complexity and runtime are all $\widetilde{O}(\sqrt{N}/\rho^2) \cdot \mathsf{poly}(\kappa)$. The protocol has 4 messages.*

- *Doubly-efficient prover: the honest prover's sample complexity is $\widetilde{O}(N) \cdot \mathsf{poly}(1/\rho)$ and its runtime is $\mathsf{poly}(N, \kappa)$.*

For simplicity, we assume throughout that the security parameter is a small polynomial in $N$. We emphasize that the protocol achieves *tolerant* verification Parnas et al. (2006): the verifier should accept even if the distribution is not in the property, so long as it is *close to the property*. The complexity is polynomial in the gap $\rho = (\epsilon_f - \epsilon_c)$ between the distances. Tolerant verification can be used to approximately verify the distribution's distance to the property: if the prover claims the distance is $\delta$, we can verify this (up to distance $\rho$) by setting $\epsilon_c = \delta$ and $\epsilon_f = \delta + \rho$ in our proof system, from which it is possible to conclude the distance is at most $\delta + \rho$ with high probability. Similarly, setting $\epsilon_c = \delta - \rho$ and $\epsilon_f = \delta$ yields a lower bound to the distance. In the distribution testing setting (without a prover) tolerant testing is a notoriously hard problem: many well-studied problems require quasi-linear sample complexity (see below). For clarity of exposition, protocols are presented as if the honest prover has *perfect* knowledge of the distribution, but this idealized honest prover can implemented by an honest prover that learns a sufficiently-accurate approximation to the distribution.

**On the complexities.** Our protocol's sample complexity is nearly optimal: even non-tolerant verification of simple properties requires $\Omega(\sqrt{N}/\rho^2)$ samples (indeed where $\rho$ is the distance parameter). This holds regardless of the communication or round complexities Chiesa & Gur (2018); Herman & Rothblum (2022) (their proof extends to computationally sound argument systems). As noted above, for distribution testing without a prover, and especially for tolerant testing, many properties require linear or quasi-linear $\Omega(N/\log(N))$ sample complexity. For example, this is true for tolerant uniformity testing, approximating the support size or the Shannon Entropy and more Raskhodnikova et al. (2009); Valiant & Valiant (2010). Since proving is at least as hard as testing (running the prover and the verifier together gives a tester), any honest prover that seeks to prove these properties will require $\widetilde{\Omega}(N)$ sample complexity and runtime. Note that our prover runs in polynomial time and achieves quasi-linear sample complexity. This sample complexity is thus optimal up to $\mathsf{polylog}(N)$ factors and dependence on the distance parameter.

**Huge domain, bounded support.** The result of Theorem 1.1 can be extended to distributions over a huge domain $\mathcal{U}$, so long as the support size of the distribution is bounded by $N$ (we need this bound to hold in both the YES case and the NO case). The complexities all incur a poly-logarithmic overhead in $|\mathcal{U}|$. The extension follows immediately from a general domain-reduction technique of Herman and Rothblum (Herman & Rothblum, 2024, Section 7). As argued in Herman & Rothblum (2024), this is quite a natural setting: for example, it lets us verify properties of a distribution over at most $M$ individuals, even if each individual's representation can come from a rich domain.

---

[3]The setting where $\rho(N) \in ((1/\sqrt{N}), 1)$ is the main setting of interest for our problem. In addition, for $\rho < 1/\sqrt{N}$ our protocol's communication complexity is at least linear, and we might as well use the protocol where the prover sends a complete description of the distribution to the verifier (see below).

**Comparison to unconditionally sound protocols.** Chiesa and Gur Chiesa & Gur (2018) presented an unconditionally sound non-interactive protocol for general distribution properties. Their protocol, however, is not sublinear: verification requires quasi-linear communication and verification time. The sample complexity, however, is only $O(\sqrt{N}/\rho^2)$ (similar to our protocol). Subsequent work, ours included, has focused on sublinear-time verification. Herman and Rothblum Herman & Rothblum (2022; 2023) constructed sublinear and unconditionally sound protocols for the class of *label-invariant* properties with $\widetilde{O}(\sqrt{N}) \cdot \text{poly}(1/\rho)$ sample and communication complexities (see also Section 1.3). Our new protocol is only computationally sound (and assumes CRHs), but it applies to a much richer class of distribution properties. Further, the polynomial dependence on $\rho$ in the communication and the sample complexities is much better (for the sample complexity it is tight, see above). A very recent work Herman & Rothblum (2024) constructs unconditionally sound protocols for any property that can be approximately decided in bounded-depth (and polynomial size) or bounded-space (and polynomial time). Here too, our class of properties is richer (there is no depth or space restriction on the approximate decision procedure). Their protocols also have $\widetilde{O}(N^{1-\alpha}) \cdot \text{poly}(1/\rho)$ sample and communication complexities for a constant $\alpha < 0.1$: i.e. the gaps with the complexity in our work are even larger.

**About public-coin protocols for distribution properties.** The protocol behind Theorem 1.1 requires the verifier to draw samples from the distribution $D$, and send them to the prover, thus preventing it from being a public-coin protocol, in which the verifier only sends random bits to the prover. It remains open whether this protocol can be made public-coin. We note public-coin protocols allow for the application of many powerful tools - one example of particular interest is the Fiat-Shamir heuristic, through which a public-coin protocol can be made non-interactive, see Fiat & Shamir (1986). For a deeper discussion on public-coin protocol for distributions as well as a construction of such protocols for a family of distribution properties, see Herman (2024).

## 1.2 EXTENSIONS AND APPLICATIONS

We discuss several applications of the protocol of Theorem 1.1 and an extension to NP properties.

**Tolerant verification for monotone distributions, juntas and more.** The protocol of Theorem 1.1 can be used for *tolerant* verification of several properties that are not label-invariant, and where known testers have prohibitive sample complexity:

1. A distribution over $\{0,1\}^n$ is *monotone* if flipping the value of a coordinate from zero to one can only increase the probability of an element. Distribution testing and learning problems for monotone distributions were studied in Batu et al. (2004); Rubinfeld & Servedio (2005); Rubinfeld & Vasilyan (2020). Taking $N = 2^n$, our protocol allows a verifier to verify a distribution's distance from being monotone using $\widetilde{O}(\sqrt{N}/\varepsilon^2)$ samples and communication. This is in contrast to the situation for distribution testing: the best tester known for separating the case where the distribution is monotone from the case where it is $\varepsilon$-far from monotone uses $O(N/(\log(N) \cdot \varepsilon^2))$ samples Rubinfeld & Vasilyan (2020).

2. A distribution over $\{0,1\}^n$ is a *k-junta* if there is a set of at most $k$ features that determine the probabilities of all elements (elements that are identical on those features have identical probabilities). Taking $N = 2^n$, Aliakbarpour, Blais and Rubinfeld Aliakbarpour et al. (2016) showed a test using $\widetilde{O}(\sqrt{N} \cdot k)$ samples. Our protocol can be used to *verify* a distribution's distance from being a $k$-junta (i.e. tolerant verification) using $\widetilde{O}(\sqrt{N}/\varepsilon^2)$ samples and communication. Our protocol also applies to the setting of a non-uniform underlying distribution (see Aliakbarpour et al. (2016)). To the best of our knowledge, no testers with less than quasi-linear sample complexity are known for these tasks.

3. Canonne *et al.* Canonne et al. (2018) studied several properties of discrete distributions over $[N]$. These include log-concavity, convexity, concavity, monotone hazard rate, unimodality and $t$-modality. They showed that tolerant testing for all these properties requires quasi-linear sample complexity in $N$. Our protocol can be used for tolerant verification of all these properties (and other properties, such as piecewise-polynomial) using $\widetilde{O}(\sqrt{N}/\varepsilon^2)$ samples and communication. We remark that we use the fact that for all properties men-

tioned above, given an explicit description of a distribution, there is a poly($N$)-time algorithm for approximating its distance from the property.

**Verifying ERM learning algorithms.**   Similarly to an application described by Herman & Rothblum (2024), our protocol can also be used in the following supervised learning setting. An untrusted data analyst claims that a hypothesis $h$ was produced by a polynomial-time empirical risk minimization algorithm $\mathcal{M}$ on a dataset $X$ of $N$ labeled examples. The argument system of Theorem 1.1 can be used to verify that $h$ is an approximate risk minimizer, where the verifier only needs to draw $\widetilde{O}(\sqrt{N}/\varepsilon^2)$ samples from the dataset $X$. If the size of the dataset is roughly the VC dimension of the benchmark class $\mathcal{H}$ (the optimal sample complexity for agnostic learning), then the verification is sublinear in the VC dimension: provably more efficient than learning would be! The main novelty here is that our protocol extends the Herman & Rothblum (2024) result to any polynomial-time ERM algorithm (their result applied to bounded-space or bounded-depth algorithms) and improves the verifier's sample and communication complexities. We rely, however, on cryptographic assumptions.

**Properties in** NP**.**   The result of Theorem 1.1 applies to an even richer class of distribution properties. Given the full explicit description of a distribution (an input of length $\widetilde{O}(N)$), we need the problem of approximating the distribution's distance to the property to be in NP (or rather in FNP, if we view this as a search problem). The theorem holds as stated for any such property, except that the honest prover needs to have a witness to the distance between the property and (a good approximation to) the distribution. For example, this can be useful in proving properties of a distribution over encrypted data, if the prover knows the users' secret keys.

## 1.3   AN EFFICIENT VERIFIED DISTRIBUTION-ORACLE

The construction behind Theorem 1.1 utilizes a lightweight and efficient sub-protocol that is of independent interest, and can be used to obtain direct and efficient protocols for a rich subclass of distribution properties. Informally, the *verified distribution-oracle protocol* is a computationally sound protocol, where the prover *commits* to a distribution $Q$ over $[N]$ using a succinct commitment. The verifier, who has sampling access to an unknown distribution $D$, can repeatedly query several functionalities for the committed distribution $Q$, and the protocol guarantees that the prover's answers are all according to a single fixed distribution $Q$, and that this distribution is $\varepsilon$-close to the unknown distribution $D$ (otherwise the verifier rejects w.h.p.). The functionalities that the verifier can query include:

1. The probability of any queried element $x \in [N]$.
2. The cumulative mass (the cdf) of all elements up to and including $x$ (according to the natual ordering of elements in $[N]$).
3. The distribution's *quantile function*, which maps an input $\mu \in (0, 1]$ to the smallest element $x$ whose cdf is $\mu$ or larger.
4. Obtaining a sample drawn by $Q$.

The sample complexity, communication and verifier runtime in the protocol are $\widetilde{O}(\sqrt{N})/\varepsilon^2$, where each query can be performed using an additional poly$(\log(N), \kappa)$ communication (and no further samples from $D$).

The verified distribution-oracle protocol lets us translate a sublinear-time algorithm *that has access to the powerful functionalities* described above w.r.t. an unknown distribution and decides a property $\Pi$, into an argument system for verifying $\Pi$. The point is that the verifier in the argument system *only has sampling access* to the unknown distribution $D$. The protocol is lightweight and efficient in the sense that it only uses simple distribution testing tools and cryptographic hash trees, whereas the full protocol of Theorem 1.1 uses PCP machinery (see below). In particular, under reasonable assumptions about the complexity of approximately deciding membership in the property, the honest prover in the lightweight protocol runs in $\widetilde{O}(N) \cdot \text{poly}(\kappa, 1/\rho)$ time.

One direct application of the verified distribution-oracle protocol is a lightweight argument system for *label-invariant* distribution properties (sometimes referred to as a *symmetric* properties):

properties where changing the labels of elements in the support of a distribution does not change membership in (or distance from) the property (see Appendix A.1). A distribution's distance from a label-invariant property only depends on the distribution's probability histogram: the number of elements with probability $p$ for each $p \in (0, 1]$ (up to an appropriate discretization of probabilities). The verifier can use the verified distribution-oracle protocol to obtain a very good approximation for $Q$'s probability histogram by drawing samples from $Q$ together with their probabilities. It can then estimate $Q$'s distance from the property, and since $Q$ is guaranteed to be close to $D$, this gives a good approximation to $D$'s distance. Herman and Rothblum Herman & Rothblum (2022; 2023) constructed unconditionally sound protocols for label-invariant properties, but their dependence on the distance parameter $\rho$ was a much larger polynomial.

## 1.4 Further Related Work

We study the verification of distribution properties via computationally sound interactive argument systems. Unconditionally sound interactive proof systems were introduced by Goldwasser, Micali and Rackoff Goldwasser et al. (1985) in the context of proving computational statements about an input that is fully known to the prover and the verifier. Computationally sound arguments were introduced by Brassard et al. (1988) in a similar context. In our work, the distribution can be thought of as the input, but it is not fully known to the verifier. We aim for verification without examining the distribution in its entirety, using minimal resources (samples, communication, runtime, etc.). Our work builds on a line of work that studies the power of sublinear time verifiers, who have query access to an unknown string, in unconditionally sound proof systems Ergün et al. (2004); Rothblum et al. (2013); Gur & Rothblum (2018) and in computationally sound arguments Kalai & Rothblum (2015). Our results are most closely related to the works on unconditionally sound protocols for verifying properties of distributions, where the verifier has sampling access to an unknown distribution (rather than query access to a string) Chiesa & Gur (2018); Herman & Rothblum (2022; 2023; 2024), and on verifying the result of machine learning algorithms using a small number of labeled examples Goldwasser et al. (2021); Gur et al. (2024). We differ from prior work on verifying properties of distributions in considering the relaxation to computational soundness. Recently Bell et al. (2024), also using cryptographic assumptions and tools, considered a setting where both the verifier and prover have *full information* about some distribution, and the verifier needs to be convinced that a value (hidden inside a commitment) was indeed drawn from the known distribution. This is in contrast to our work, where the verifier doesn't know the distribution but can only sample from it. Our distribution-commitments allow an untrusted prover to commit to the entire description of the *unknown* distribution (rather than a single unknown sample from a known distribution).

## 2 Technical Overview of Our Protocol

**Background.** Our protocol builds on *identity testers* from the distribution testing literature Batu et al. (2001). An identity tester is given an explicit description of a distribution $Q$ over a domain $[N]$: e.g. a table detailing the precise probability of every element in the domain. The tester is also given sample access to an unknown distribution $D$ over the domain $[N]$. If $D \equiv Q$ then the tester should accept, but if the statistical distance between $D$ and $Q$ is $\varepsilon$ or larger, then the tester should reject w.h.p. We emphasize that this problem is in the standalone distribution testing model (i.e. there is no untrusted prover). The sample complexity of identity testing is $\Theta(\sqrt{N}/\varepsilon^2)$ Valiant & Valiant (2014).

Chiesa and Gur Chiesa & Gur (2018) suggested a natural protocol that uses identity testers to *verify* general distribution properties: the untrusted prover sends a complete description of a distribution $Q$, and alleges that this describes the unknown distribution $D$. The verifier, who doesn't trust the prover, uses an identity tester to verify that $Q$ is not far from $D$. If this is the case, then $Q$ can be used as a surrogate for $D$ in determining membership in (or distance from) the property. Further, since $Q$ is fully specified, the verifier can decide its membership in (or distance from) any efficiently decidable property: this is a purely computational problem and no additional samples from $D$ are needed.

The issue is that sending the description of the distribution requires linear communication (to specify the probability of each element), whereas we are interested in protocols with sublinear communication complexity and verification time. Moreover, the verifier needs to run the procedure that decides,

given $Q$'s description, whether the distribution is in the property. For the general class of properties we study, this latter test can require arbitrary polynomial runtime (even the linear time needed to "read" the distribution description is more than we can afford).

**What does the identity tester need to know about $Q$?**   Identity testers use samples drawn from the unknown distribution $D$, together with information about the known distribution $Q$. A key question in our work is exactly what the identity tester needs to know about $Q$. Focusing on an identity tester of Goldreich (Goldreich, 2017, Section 11.2.2) with optimal $O(\sqrt{N}/\varepsilon^2)$ sample complexity, we show it only needs access to an oracle that, for an element $x \in [N]$, provides the probability $Q[x]$, together with the ability to draw samples from $Q$.

We focus on the requirement of giving the tester access to an oracle that answers queries $x \in [N]$ with $Q[x]$ (we ignore the issue of sampling from $Q$ for now). We might consider the following naive protocol: the verifier runs the identity tester and asks the untrusted prover to specify the probability $Q$ assigns to elements the tester queries, and accepts iff the identity tester accepts. The problem with this naive approach is that a cheating prover can answer the verifier's queries in a way that is not consistent with any global distribution $Q$. There are two issues: first, the cheating prover can be adaptive, so the probability that it specifies for an element $x$ can depend on the other elements queried by the verifier. Second, even if the prover were non-adaptive and answered each query $x$ using a fixed probability determined in advance, there is no guarantee that the different probabilities specified for each element specify a probability distribution, i.e. that they sum to 1, and this can lead to the identity tester failing. How can we ensure that the untrusted prover answers the verifier's queries using a fixed and valid underlying distribution without having it send (and commit to) a complete description of the distribution?

## 2.1 SUCCINCT DISTRIBUTION-COMMITMENTS WITH LOCAL OPENINGS

We use cryptographic hashing to design a scheme by which the untrusted prover first commits to the entire distribution by sending a short digest (much shorter than the description of the entire distribution). For the honest prover, it can send a digest that allows it to *verifiably open* the probability of any element $x \in [N]$. The opening includes a proof of correctness, and it is *local* in the sense that verifying its validity only requires $\text{poly}(\log(N), \kappa)$ communication and computation. For security, the commitment is *binding*: for any polynomial-size cheating prover, once it sends a digest, it can only successfully open the probabilities of elements according to a fixed underlying distribution $\widetilde{Q}$. Importantly, $\widetilde{Q}$ is completely determined by the digest and is guaranteed to be a valid probability distribution (the probabilities of all elements sum up to 1).

We construct such distribution-commitments using collision-resistant hash functions: families of shrinking hash functions where, given a random function in the family, it is hard to find a pair of inputs mapped to the same output (a collision). The verifier chooses the hash function and sends it to the prover. The prover constructs a binary hash tree with $N$ leaves, where for $x \in [N]$ the $x$-th leaf is labeled with the probability $Q[x]$. We define the probability of an internal node to be the sum of the probabilities of its children (equivalently, the probability of an internal node is the sum of probabilities of leaves in its sub-tree). The label of each internal node in the tree includes its probability and a hash of the labels of its children. The digest is the label of the root of the tree (which should have probability 1). The local opening for $x \in [N]$ includes all the labels of nodes on the path from the root to $x$ and of their siblings. The verifier verifies consistency of the hash values and of the probabilities (for every internal node on the path, its probability is the sum of probabilities of its children). Collision-resistance ensures that the digest binds the prover to at most one label that it can successfully open for each node in the tree. In particular, for each leaf $x \in [N]$ there is (at most) only one probability $\widetilde{Q}[x]$ that the prover can open. We also show that the probabilities of leaves that the prover can successfully open are consistent with a global distribution. This latter security guarantee is formalized using an extraction-based definition: given the digest and black-box access to a cheating prover, we can extract $\widetilde{Q}$ efficiently.

We remark that the prover might refuse to open some locations in the hash tree, or open them in a way that is not consistent with $\widetilde{Q}$ and is immediately rejected. The point, however, is that any query that is met with a refusal to open or an inconsistency leads to immediate rejection. Thus, when we use the dsitribution-commitment in an argument system, the prover's success probability can be

upper-bounded by its probability in a mental experiment where all queries are answered using $\widetilde{Q}$. Thus, for simplicity and w.l.o.g., in the techincal overview we assume that the prover answers all queries according to $\widetilde{Q}$. See Appendix B for further details and discussion.

We note that distribution-commitments draw inspiration from the cryptographic literature on accumulators Benaloh & de Mare (1993); Baric & Pfitzmann (1997) and a construction of Buldas et al. (2000). An accumulator is a commitment to a set of values, whereas we need the commitment to specify probabilites for the values, and for these probabilities to sum to 1.

**Recovering the cdf and sampling from $Q$.** Our protocols also need the ability to locally open, for any $x \in [N]$, the cumulative probability of mass $Q$ assigns to the elements up to (and including) $x$. We denote this cdf by $\Phi_Q(x)$. The hash tree construction supports this functionality using the same opening: the verifier sums the probability of $x$ with the probability of all left-siblings of internal nodes on the path from $x$ to the root. Access to the cdf function is important for our protocol for general polynomial-time properties. It is also useful for supporting sampling from $Q$ (which is used by the identity tester, see above): to sample from $Q$ the verifier picks a random probability $\mu \in [0, 1]$ and asks the prover to open the smallest element $x$ whose *cdf* is at least $\mu$ (the *quantile* function). See further details in Appendix B.

## 2.2 VERIFIABLE DISTRIBUTION-ORACLES

In our protocol, the untrusted prover commits to $Q$ using the distribution-commitment. The verifier then runs an identity tester, drawing samples from the unknown distribution $D$, and using the local opening to obtain all the information it needs about $Q$ for each of its samples. The main point about this protocol is that the communication is much smaller than the complete description of $Q$: the digest is short, and each opening is of $\mathsf{polylog}(N)$ length. The identity tester uses $O(\sqrt{N}/\varepsilon^2)$ samples, so the total communication and verification time are $\widetilde{O}(\sqrt{N}/\varepsilon^2)$.

The identity tester guarantees that $Q$ is close to the unknown distribution $D$ (otherwise the verifier rejects w.h.p.). The distribution-commitment scheme allows the verifier to recover the proability and the cdf of any element $x \in [N]$, to answer quantile queries, and to sample form $Q$ (see above). Thus, it can be used towards direct and efficient protocols for a rich sub-class of distribution properties, including label-invariant properties. See Section 1.3.

## 2.3 VERIFYING PROPERTIES IN $\mathcal{P}$ (AND BEYOND)

Our main result is an interactive argument system for verifying *any efficiently-decidable* distribution property. We assume there is a $\mathsf{poly}(N)$-time Turing machine $\mathcal{M}$ that, given as input a complete description of a distribution $Q$ over $[N]$ (say as a list of elements and their probabilities), computes $Q$'s distance to the property $\Pi$, up to a $\tau$ additive approximation error.

**Interactive arguments of proximity (IAP).** Our construction uses IAPs Rothblum et al. (2013); Kalai & Rothblum (2015): protocols for deciding approximate membership of an input string $X \in \{0, 1\}^n$ in a language $\mathcal{L}$. The verifier has *query access* to the input. It should accept if the input is in the language. If the input is $\varepsilon$-far from the language in relative Hamming distance, then no polynomial-size cheating prover should be able to get the verifier to accept (except with negligible probability). Assuming the existence of collision-resistant hash functions, any language in $\mathcal{P}$ has an IAP that proceeds in two steps: first, in the *communication phase*, the verifier and the prover communicate in a 4-message protocol with communication complexity and verifier time $\mathsf{poly}(\log(N), \kappa)/\varepsilon$. The verifier does not access the input in the communication phase. In a subsequent query phase, the verifier makes $O(1/\varepsilon)$ queries to the input and accepts or rejects. The protocol uses the PCP theorem, which induces overheads for the prover. It may be possible to reduce this overhead using recent advances from the proof-system literature Reingold et al. (2016); Ben-Sasson et al. (2016); Ron-Zewi & Rothblum (2022). See Section A.3 for further details.

**Our protocol.** At a high level, the protocol operates as follows:

- The prover commits to the distribution $D$ via a digest $d$, using the scheme of Section 2.1.

- The prover and the verifier run the verified distribution-commitment protocol of Section 2.2 to verify that the digest $d$ commits the prover to a distribution $\widetilde{Q}$ that is $\varepsilon$-close to $D$. This requires $\widetilde{O}(\sqrt{N})/\varepsilon^2$ samples from $D$ (this is the only step where the verifier accesses $D$).

- The prover and the verifier use a scheme for representing distributions as *bit strings* and on a polynomial-time Turing machine $\mathcal{M}'$ as follows. For any distribution $D'$ over $[N]$, let its representation be $X_{D'}$ (a bit string). If $D'$ is in $\Pi$,[4] then $\mathcal{M}'$ accepts $X_{D'}$. If $D'$ is $\varepsilon$-far from the property (in total-variation distance), then $X_{D'}$ is $\Theta(\varepsilon)$ far *in (relative) Hamming distance* from any string that $\mathcal{M}'$ accepts. The prover (who knows $D$) can compute its representation $X_D$, but the verifier cannot. However, the distribution-commitment scheme allows the verifer to query the representation $X_{\widetilde{Q}}$ of the distribution $\widetilde{Q}$ that the prover committed to using the digest $d$ (more on this below).

- The prover and the verifier run an interactive argument of proximity (IAP) to check if $\mathcal{M}$ accepts $X_{\widetilde{Q}}$. We emphasize that the IAP is for languages defined over strings. This is where the verifier needs query access to $X_{\widetilde{Q}}$.

  Completeness follows directly: we focus on the soundness argument. Since $\widetilde{Q}$ and $D$ are close, $\widetilde{Q}$ must be far from the property. Thus, the representation guarantees that $X_{\widetilde{Q}}$ is far from making $\mathcal{M}'$ accept. The binding property of the commitment scheme and the soundness of the IAP guarantee that the verifier will reject w.h.p.

**String representation and consistency.** For simplicity, suppose that $D$ is $\eta$-grained for $\eta \in (0, 1/N)$ Goldreich et al. (2023). I.e., that the probabilities of all elements in $D$ are integer multiples of $\eta$. We represent the distribution $D$ as a string $X_D \in \Sigma^{(1/\eta)}$ where the alphabet $\Sigma$ is of size $\text{poly}(N)$. We first divide the mass of the distribution into $(1/\eta)$ "grains", each representing $\eta$ of the mass of the distribution. Each element $x$ with probability $D[x]$ contributes $(D[x]/\eta)$ grains that have value $x$. We then sort the grains and write them into the string $X_D$. This means that

$$\forall x \in [n], \forall i \in \left[ \frac{\Phi_D(x) - D[x]}{\eta}, \ldots, \frac{\Phi_D(x)}{\eta} \right], X_D(i) = x.$$

In turn, any sorted string corresponds to a distribution in the natural way. Given the string $X_D$, we can reconstruct the standard representation of $D$ as a list of probabilities using a simple one-pass procedure. The machine $\mathcal{M}'$ uses this procedure to decide, given $X_D$, whether $D$ is in the property. By construction, if $\widetilde{X}$ is close to $X_D$ in Hamming distance (and is sorted), then the distribution $\widetilde{D}$ that it represents is close to $D$ (the converse might not hold). The full construction uses a high-distance error-correcting encoding of each element $x$ to get a tight relationship between the Hamming distance over strings and the statistical distance over the distributions they represent.

Given a digest $d$ specifying a distribution $\widetilde{Q}$, and an index $i \in [(1/\eta)]$, the verifier can query the $i$-th symbol of $X_{\widetilde{Q}}$ by asking the prover to open the digest $d$ to reveal the smallest element whose cdf by $\widetilde{Q}$ is at least $(\eta \cdot i)$ (answering the quantile function, see Section 1.3).

ACKNOWLEDGMENTS

This project has received funding from the European Research Council (ERC) under the European Union's Horizon 2020 research and innovation programme (grant agreement No. 819702) and from the Simons Foundation Collaboration on the Theory of Algorithmic Fairness.

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

## A  NOTATIONS AND FORMAL DEFINITIONS

For an integer $n \in \mathbb{N}$, we use $[n]$ to denote the set $\{1, \ldots, n\}$. We use the natural ordering over integers to order the elements in any subset of $[n]$.

**Turing machines.**  A probabilitic polynomial-time Turing Machine (PPTM) is a Turing Machine that has access to a tape cointaining uniformly random i.i.d. coins and runs in polynomial time. A PPTM that participates in an interactive protocol is augmented with an interaction tape for sending and receiving messages. A *non-uniform* machine also gets access to a polynomial-length advice string (the advice string is fixed for each input length $n$). We sometimes refer to the running time of a non-uniform machine as its *size* (drawing on equivalences between non-uniform Boolean circuits and non-uniform Turing Machines).

An *oracle machine* with oracle access to a probabilistic functionality $\Gamma$ has an additional oracle tape where it can write a query $z$ and get back a sample from $\Gamma(z)$ at unit cost. The functionality might take empty inputs, in which the machine has sampling access to a fixed distribution $\Gamma$.

**Distributions.**  We work with distributions over discrete domains throughout this work. For a set $S$, we denote the uniform distribution over $S$ by $U_S$. For a distribution over a domain of size $N$, we assume w.l.o.g. that all probabilities are taken up to precision $\text{poly}(1/N)$, so an element's probability can be represented using $O(\log(N))$ bits. For a distribution $Q$ over $[N]$, we use $Q[x] \in [0, 1]$ to denote the probability of $x \in [N]$. Similarly, we use $\Phi_Q(x)$ to denote the *cumulative* mass of all elements up to (and including) $x$, i.e. $\Phi_Q(x) = \sum_{x' \in [N]:x' \leq x} Q[x]$.

**Definition A.1** *The total variation distance (alt. statistical distance) between distributions $P$ and $Q$ over a finite domain $X$ is defined as:*

$$\delta_{TV}(P, Q) = \frac{1}{2} \sum_{x \in X} |P(x) - Q(x)|$$

## A.1 DISTRIBUTION PROPERTIES AND TESTING

**Definition A.2 (Distribution properties)** *A distribution property $\Pi = (\Pi_N)_{N \in \mathbb{N}}$ is an ensemble of sets of distributions, where $\Pi_N$ contains distributions over the domain $[N]$.*

*The* distance *of a distribution $D$ over domain $[N]$ from distribution property $\Pi$ is*

$$\delta_{TV}(D, \Pi) \stackrel{\Delta}{=} \inf \{\delta_{TV}(D, Q) : Q \in \Pi_N\}$$

**Definition A.3 (Property tester)** *An algorithm $T$ that is given parameter $N \in \mathbb{N}$, distance parameter $\epsilon \in (0, 1)$, and samples drawn from a distribution $D$ over domain $[N]$ is called a* tester *for distribution property $\Pi$ if the following hold:*

- *If $D \in \Pi$, $T$ accepts with probability at least $2/3$.*

- *If $\delta_{TV}(D, \Pi) > \epsilon$, $T$ rejects with probability at least $2/3$.*

*The success probabilities can be amplified by repetition. The* sample complexity *is the number of samples $T$ draws from $D$ (as a function of $N$ and $\epsilon$).*

**Theorem A.4 (Uniformity tester, Goldreich & Ron (2000); Paninski (2008))** *The sample complexity for testing the property of being uniform over the entire domain, i.e. $\mathcal{U} = (\{U_{[N]}\})$, is $\Theta\left(\sqrt{N}/\epsilon^2\right)$.*

**Definition A.5 (Label invariant distribution properties)** *We call a distribution property $\Pi$ label-invariant if for every $D$ over $[N]$ and for every permutation $\pi : [N] \to [N]$ it holds that:*

$$D \in \Pi \Leftrightarrow \pi(D) \in \Pi.$$

The family of label-invariant distribution properties contains many well-studied and natural properties, for example: the property $\mathcal{U}$ of being uniform over the domain (see Theorem A.4), the property of having entropy $K(N)$, the property of having support of size at most $S(N)$, the property of being at distance at most $\delta(N)$ from a uniform distribution over a subset of the domain, etc.

**Efficiently decidable properties.** We define two families of distribution properties where questions like membership in or distance from the property can be decided efficiently given an appropriate description of the distribution. First, we define label-invariant properties where membership can be decided in polylogarithmic time given the approximate histogram.

**Definition A.6 (polylog-time approximately decidable label-invariant properties)** *A label-invariant property $\Pi$ is polylog-time $\tau$-approximately decidable if the following two conditions hold.*

- *There exists an algorithm $\mathcal{A}_{decide}$, that is given parameters $\tau \in (0, 1)$, $N \in \mathbb{N}$, and a $\tau$-approximate probability histogram $(p_j)_j$ and runs in $\mathsf{poly}(\log(N), 1/\tau)$ time. If there exists a distribution $Q$ over domain $[N]$ with $\tau$-approximate histogram $(p_j)_j$ such that $\delta_{TV}(Q, \Pi_N) \leq \tau$, the algorithm accepts; and if every distribution consistent with the given histogram satisfies $\delta_{TV}(Q, \Pi_N) > 2\tau$, it rejects.*

- *There exists an algorithm $\mathcal{A}_{find}$, that is given parameters $N \in \mathbb{N}$ and $\delta, \rho \in (0, 1)$, as well as an explicit description of a distribution $D$ over domain $[N]$ such that $\delta_{TV}(D, \Pi_N) \leq \delta$. $\mathcal{A}_{find}$ runs in time $\widetilde{O}(N)\mathsf{poly}(\delta^{-1}, \rho^{-1})$, and outputs an explicit distribution $Q$ over domain $[N]$ such that: $Q \in \Pi_N$ and $\delta_{TV}(D, Q) \leq \delta + \rho$.*

**Remark A.7** *We view the condition that a property admits the two algorithms described in Definition A.6 as mild. The* decide *algorithm requires that a membership of a distribution in a property can be approximated efficiently just given its histogram. Herman and Rothblum (Herman & Rothblum, 2022, Section 6.2) present* decide *algorithms for fundamental label-invariant properties: e.g. for approximating the Shannon entropy and the distance from uniform. The* find *algorithm is also implementable for those properties and many more.*

**Definition A.8** *[Efficient approximately decidable properties] For $\rho(N) \in (0, 1)$, a distribution $\Pi$ is* poly-*time (or efficiently) $\rho(N)$-approximately decidable if it admits a* poly$(N, 1/\rho)$-time algorithm $\mathcal{D}ist$ that receives as input $N$ and an explicit description of a distribution $D$ over domain $[N]$ and outputs $\delta$ such that:

$$|\delta_{TV}(D, \Pi_N) - \delta| \leq \rho(N)$$

## A.2 CRYPTOGRAPHIC DEFINITIONS

A function $f : \mathbb{N} \to \mathbb{N}$ is polynomial-time if there is a poly$(\kappa)$-time Turing Machine that on input $1^n$ outputs $f(n)$. A function $q : \mathbb{N} \to [0, 1]$ is *negligible* if for any polynomial $p(n)$, for all but finitely many input lengths, $f(n) < 1/p(n)$. We denote this by $q(n) = \mathrm{negl}(n)$.

In the cryptographic definitions in this work we focus on non-uniform polynomial time adversaries. In our setting, it is often the case that there is an unknown underlying distribution $D$ over a domain $[N]$, and the adversary has sampling access to $D$. We remark that any such adversary can be converted into a non-uniform adversary with no access to $D$ by a standard argument (plugging a "good" collection of samples into the adversary's non-uniform advice string). This transformation applies even when the domain of the distribution is exponential in the adversary's running time, but in this work we mostly focus on adversaries whose runtime is polynomial in the domain size.

**Collision-Resistant Hash Families (CRHs).** CRHs are families of length-shrinking functions where, given the description of a random function $h$ in the family, it is hard for a PPTM adversary to find a collision, i.e. $x \neq x'$ in $h$'s domain s.t. $h(x) = h(x')$.

**Definition A.9 (CRH)** *Let $\ell_{\mathrm{in}}, \ell_{\mathrm{out}} : \mathbb{N} \to \mathbb{N}$ be* poly$(\kappa)$-*time computable sequences of input and output lengths, where $\forall \kappa \in \mathbb{N}, \ell_{\mathrm{out}}(\kappa) < \ell_{\mathrm{in}}(\kappa)$. A family of collision resistant hash functions (CRHs) is defined by PPTM algorithms $\mathcal{G}en$ and $\mathcal{E}val$:*

- *$\mathcal{G}en(1^\kappa)$ outputs* key*, where $h_{\mathrm{key}} : \{0, 1\}^* \to \{0, 1\}^{\ell(\kappa)}$ is a function in the family $H_\kappa$.*

- *$\mathcal{E}val(\mathrm{key}, x \in \{0, 1\}^*)$ outputs $h_{\mathrm{key}}(x)$.*

*For any polynomial-size $\mathcal{A}dv$, the probability that it finds a collision is negligible:*

$$\Pr_{\mathrm{key} \leftarrow \mathcal{G}en(1^\kappa)} [(x, x') \leftarrow \mathcal{A}dv(1^\kappa, \mathrm{key}), \ x \neq x', \ h_{\mathrm{key}}(x) = h_{\mathrm{key}}(x')] = \mathrm{negl}(\kappa).$$

Recall that we allow $\mathcal{A}dv$ to be non-uniform. Namely, we assume that the CRHs are secure against non-uniform adversaries. Note also that the existence of CRHs where $\ell_{\mathrm{out}} < \ell_{\mathrm{in}}$ is equivalent to the existence of CRHs with arbitrary polynomial shrinkage. Finally, we note that throughout this work we typically assume that the security parameter is (at least) a small polynomial in the domain size $N$ of the unknown distribution.

## A.3 INTERACTIVE ARGUMENTS OF PROXIMITY (IAPs)

IAPs were informally described in Rothblum et al. (2013) and formally studied by Kalai and Rothblum Kalai & Rothblum (2015). They are computationally-sound variants of Interactive Proofs of Proximity (IPPs), which were defined and studied in Ergün et al. (2004); Rothblum et al. (2013). These are interactive protocols for proving membership of an input *string* in a language. The verifier has query access to the input, and should run in sublinear time in the input length. In an IAP, no polynomial-time cheating prover should be able to get the verifier to accept any input that is *far* from the language (except with negligible probability). The main difference from arguments for *distribution properties* (our focus in this work, see Section A.4), is that the input to the proof-system is a string, rather than a distribution, and the verifier has *query access* to this string, rather than sampling access in the distribution case.

For our definition, we consider an IAP that proceeds in two phases: a *communication phase*, where the prover and the verifier exchange messages, but the verifier does not query the input, and a subsequent *query phase*, where the verifier queries the input. The verifier then applies a *decision predicate* to its views in the communication phase and the query phase (including any random coins it tossed), and rejects or accepts. See Goldreich et al. (2023) for a more thorough structural study of sublinear-time verifiers in proofs of proximity for strings.

**Definition A.10 (Interactive Argument of Proximity (IAP) Rothblum et al. (2013); Kalai & Rothblum (2015))**
*An IAP for a language $\mathcal{L}$ is an interactive protocol with two parties: a prover $\mathcal{P}$ and a verifier $\mathcal{V}$. Both parties get an input length $n$, a proximity parameter $\epsilon \in (0,1)$ and a secruity parameter $1^\kappa$. The verifier also gets oracle access to an input $x \in \{0,1\}^n$, whereas the prover has full access to $x$.*

*The protocol is divided into two phases. In the* interaction phase *the two parties interact, but the verifier does not access the implicit input. The interaction produces a communication transcript $\tau$. In the subsequent* query phase*, the verifier makes its queries $Q$ into the implicit input (based on the explicit input, the transcript $\tau$, and its random coins $r$). The verifier then applies a decision predicate $\phi_\mathcal{V}$ to its views from both phases and accepts or rejects.*

1. **Completeness:** *For every input $x \in \mathcal{L}$ and $\epsilon > 0$ it holds that*

$$\Pr_{(r,\tau,Q)\leftarrow\left(\mathcal{P}(x),\mathcal{V}^x\right)(|x|,\epsilon,1^\kappa)}\left[\phi_\mathcal{V}(r,\tau,(x|Q))=1\right]=1.$$

2. **Soundness:** *For every $\epsilon > 0$ and $x$ that is $\epsilon$-far from the language $\mathcal{L}$, and for every $\mathsf{poly}(\kappa,|x|)$-time non-uniform cheating prover $\widetilde{P}$ it holds that*

$$\Pr_{(r,\tau,Q)\leftarrow\left(\mathcal{P}^*(x),\mathcal{V}^x\right)(|x|,\epsilon,1^\kappa)}\left[\phi_\mathcal{V}(r,\tau,(x|Q))=1\right]=\mathsf{negl}(\kappa).$$

We typically measure the complexity of an IAP in terms of the verifier's query complexity, the communication complexity, the number of rounds, and the prover's and the verfier's runtimes (the verifier's runtime is taken over both phases). We note that the division of the protocol into communication and query phases is (to a large extent) without loss of generality: even if the verifier needs to know values of the input on-the-fly during the communication phase, it can ask the prover for the values the input takes on those queries, and then check the prover's answers in the query phase. As we do throughout this work, we typically assume that the security parameter is at least a small polynomial in the input length $n$.

**Theorem A.11 (IAPs for NP Rothblum et al. (2013))** *Let $\mathcal{L}$ be an NP language, $\varepsilon = \varepsilon(n)$ a proximity parameter and $\kappa = \kappa(n)$ a security parameter. Assuming the existence of collision-resistant hash functions (see Definition A.9), $\mathcal{L}$ has a public-coin 4-message IAP for $\varepsilon$-proximity with query complexity $O(1/\varepsilon)$. The communication complexity and verifier runtime are $(\mathsf{poly}(\log(n),\kappa)/\varepsilon)$. The honest prover, given a witness to the input $x$'s membership in $\mathcal{L}$, runs in $\mathsf{poly}(n,\kappa)$ time.*

The protocol also applies to languages in $\mathcal{P}$, where the witness for membership is empty: the honest prover runs in polynomial time given only the input, with no additional auxiliary information. The construction builds on Kilian's 4-message argument system for NP languages. The idea is for the prover to commit to a locally testable and locally decodable encoding of the input, as well as PCP of proximity Ben-Sasson et al. (2006) for the input's membership in the language. The commitment should be a succinct and locally-openable string commitment, which can be based on any CRH family (e.g. using a hash tree). The verifier in the argument system gets the commitment, runs the PCPP verifier, and asks the prover to open its queries into the proof and the input encoding. It also checks that the committed input encoding is (close to) a valid encoding of a string $x'$ that is $\varepsilon$-close to the input $x$. This can be done by locally testing the committed encoding and locally decoding $O(1/\varepsilon)$ random locations in the message inside the encoding (the local testing and decoding use the local opening of the commitment scheme). The values of the decoded locations in the committed message are compared to the actual input. This latter check is the only place where the argument's verifier accesses the input, and is performed during the query phase.

Finally, we note that Rothblum et al. (2013) suggested that the protocol could be made non-interactive using the Fiat-Shamir heuristic Fiat & Shamir (1986), as proposed by Micali Micali (1994). Soundness would be heuristic, or could be proved in the random oracle model. Kalai and Rothblum Kalai & Rothblum (2015) showed a 2-message protocol for languages in $\mathcal{P}$ based on a stronger cryptographic assumption.

## A.4 Interactive Arguments for Distribution Properties

We consider interactive protocols consisting of a prover and a verifier $(\mathcal{P}, \mathcal{V})$, where the parties send messages back and forth (in our protocol the first message is sent by the verifier). In an argument system for a distribution property, the prover and the verifier both get black-box sampling access to an unknown distribution $D$. The protocol's prover and verifier sample complexities are the total number of samples drawn by each of the parties. The protocol's communication complexity (the total number of bits sent), round complexity (the total number of messages sent by both parties), and the total runtimes of both parties are defined in the natural way.

**Definition A.12 (Interactive argument system for tolerant distribution testing)** *A tolerant interactive argument system for a distribution property $\Pi$ is an interactive protocol $(\mathcal{P}, \mathcal{V})$ where the prover and the verifier both get as explicit input the domain size $N$, the (unary encoding of the) security parameter $1^\kappa$, and proximity parameters $\epsilon_c \leq \epsilon_f$. The prover and the verifier also both get black-box (oracle) sampling access to a known distribution $D$. For every $N, \kappa \in \mathbb{N}$ and $\epsilon_c \leq \epsilon_f \in (0, 1)$, for every distribution $D$ over the domain $[N]$:*

- ***Completeness:*** *if $\delta_{TV}(D, \Pi_N) \leq \epsilon_c$, the verifier $V$, interacting with the prover $P$, accepts with all but $\mathrm{negl}(\kappa)$ probability.*

- ***Soundness:*** *if $\delta_{TV}(D, \Pi_N) \geq \epsilon_f$, then for every polynomial-size non-uniform cheating prover strategy $\widetilde{P}$, the verifier $V$, interacting with $\widetilde{P}$, rejects with all but $\mathrm{negl}(\kappa)$ probability.*

*We typically measure the runtime of the verifier and the (honest) prover as a function of $N, \kappa, \rho = (\epsilon_f - \epsilon_c)$. We aim for the honest prover to be (at most) polynomial in these parameters. The verifier is typically required to be sublinear in $N$. Throughout this work we allow the cheating prover to run in $\mathrm{poly}(N, \kappa, 1/\rho)$ non-uniform time.[5]*

For simplicity, we assume throughout this manuscript that $N$ and $\kappa$ are polynomially related.

## B Succinct Distribution-Commitments

A distribution-commitment allows sender to commit to a distribution $Q$ over a domain $[N]$ using a succinct *digest* $d$. The digest can later be used by a receiver (or verifier) to "locally" open the probability that $Q$ assigns to any element $x \in [N]$ in a verifiable way (see below). Moreover, the commitment scheme we construct also supports opening the cumulative mass $\Phi_Q(x)$ of elements up to (and including) $x$ using the natural lexicographic ordering over elements in $[N]$. We refer to these probabilities as the pdf and the cdf of $x$ (respectively). For security, we require that no PPTM adversary can produce a digest that allows it to locally open the probabilities of an element in two different ways. Moreover, successful openings are always consistent with a (unique) distribution that is specified by the digest. Of course, an adversary might send a digest and then later refuse to open the probability of a particular element (or a set of elements), but we require that once a digest $d$ has been sent, there is a unique and well-defined distribution $\widetilde{Q}$ s.t. the adversary can only get the receiver to accept probabilities according to $\widetilde{Q}$. We emphasize that this is the case even though the digest is short (its size depends only on the security parameter, independent of $N$), whereas explicitly describing the distribution could require communicating $\widetilde{O}(N)$ bits.

**Efficient extraction.** An adversarially produced digest induces a distribution $\widetilde{Q}$. We require that, once the digest is specified, the distribution $\widetilde{Q}$ can "extracted" in polynomial time by an extraction algorithm. The extractor gets black-box access to an adversary who produces a sequence of elements and opens their probabilities. Given any such PPTM adversary, the extractor produces a full description of a well-specified distribution $\widetilde{Q}$, and the adversary cannot successfully open the probabilities (the pdf and the cdf) of any element $x$ except to $\widetilde{Q}[x]$ and $\Phi_{\widetilde{Q}}(x)$ (respectively).

---

[5]More generally, it is also interesting to consider huge domains, i.e. much larger than $\mathrm{poly}(\kappa)$. In this case, one can consider $\mathrm{poly}(\kappa)$-time adversaries, whose running time is sublinear in the domain size.

**Definition B.1 (Distribution-Commitment)** *A distribution-commitment scheme has PPTMs $\mathcal{G}en$, $\mathcal{D}ig$, $\mathcal{O}pen$, $\mathcal{V}_{\mathcal{C}ommit}$ and $\mathcal{E}xt$ as follows.*

- *$\mathcal{G}en(1^{\kappa}, N)$ outputs key.*

- *$\mathcal{D}ig(\text{key}, Q)$ gets as input a key generated by $\mathcal{G}en$ and a description of a distribution $Q$ over $[N]$. It outputs a short digest $d$ of length $\text{poly}(\kappa)$ and auxiliary information $\text{aux}$ to be used in generating proofs ($\text{aux}$ can be of length $\text{poly}(\kappa, N)$).*

- *$\mathcal{O}pen(x, \text{key}, d, \text{aux})$, on input $x \in [N]$, a digest and auxiliary information produced by $\mathcal{D}ig$, outputs $Q[x], \Phi_Q(x) \in [0, 1]$ and a proof $\pi$ of length $\text{poly}(\kappa, \log(N))$.*

- *$\mathcal{V}_{\mathcal{C}ommit}(x, p, q, \text{key}, d, \pi)$, gets as input an element $x \in [N]$ and its alleged pdf and cdf $p, q \in [0, 1]$ and a proof $\pi$. Accepts or rejects the proof and runs in $\text{poly}(\kappa, \log(N))$ time.*

- *$\mathcal{E}xt^{\mathcal{A}dv}(\text{key}, 1^N, d, 1^\rho)$ gets black-box access to an adversary algorithm $\mathcal{A}dv$ with alleged (polynomial) success probability $\rho \in (0, 1)$ and outputs a description $\widetilde{Q}$ of a distribution over $[N]$.*

*An honest sender who commits to a distribution $Q$ using $\mathcal{D}ig$ can open the probabilities $p = Q[x], q = \Phi_Q(x)$ of any element $x \in [N]$, producing a proof that will (always) be accepted by $\mathcal{V}_{\mathcal{C}ommit}$.*

*For security, we consider a PPTM adversary $\mathcal{A}dv = (\mathcal{A}dv_1, \mathcal{A}dv_2)$ that operates in two steps (the first step outputs a state that is used by the adversary in the second step). In the first step, $\mathcal{A}dv_1$ gets a key and produces a digest $d$. In the second step, $\mathcal{A}dv_2$ produces a sequence of elements and their openings. The adversary's goal is producing a digest where it can open probabilities that are not consistent with a single distribution $\widetilde{Q}$. We require that for every PPTM $\mathcal{A}dv = (\mathcal{A}dv_1, \mathcal{A}dv_2)$, for every (poly-time constructible) polynomial $\eta : \mathbb{N} \to \mathbb{N}$*

$$
\begin{aligned}
\Pr\Big[ &\text{key} \leftarrow \mathcal{G}en(1^{\kappa}, N), \\
&(d, \text{state}) \leftarrow \mathcal{A}dv_1(\text{key}), \\
&\widetilde{Q} \leftarrow \mathcal{E}xt^{\mathcal{A}dv_2(\text{state})}(\text{key}, 1^n, d, 1^{\eta(\kappa)}), \\
&((x_1, \widetilde{p_1}, \widetilde{q_1}, \widetilde{\pi_1}), \ldots, (x_m, \widetilde{p_m}, \widetilde{q_m}, \widetilde{\pi_m})) \leftarrow \mathcal{A}dv_2(\text{state}), \\
&\exists i^* \in [m] \text{ s.t. } \left( (\widetilde{p_{i^*}} \neq \widetilde{Q}[x_{i^*}]) \bigvee (\widetilde{q_{i^*}} \neq \Phi_{\widetilde{Q}}(x_{i^*})) \right), \\
&\mathcal{V}_{\mathcal{C}ommit}(x_{i^*}, \widetilde{p_{i^*}}, \widetilde{q_{i^*}}, \text{key}, d, \widetilde{\pi_{i^*}}) = accept \Big] < \frac{1}{\eta(\kappa)}.
\end{aligned}
$$

We remark that the extractor needs to know an explicit bound $\eta(\kappa)$ on the success probability of the adversary, because the adversary can refuse to produce openings with all but $\eta$ probability. The extractor needs to run it enough times to see the low-probability opening event. The construction of distribution-commitments from CRHs is in Theorem B.4.

**Sampling from $Q$.** Any distribution-commitment scheme also allows for sampling from the committed distribution $Q$ as follows. The receiver chooses a random mass $\mu \in (0, 1]$ and the sender opens the smallest $x$ s.t. $\Phi_Q[x] \geq \alpha$ (using the commitment scheme's $\mathcal{O}pen$ procedure).

**Definition B.2 (Quantile function)** *For a distribution-commitment scheme, $Quantile(\mu, \text{key}, d, \text{aux})$ gets as input a (cumulative) mass $\mu \in (0, 1]$. It outputs the smallest $x \in [N]$ s.t. $\Phi_Q(x) \geq \mu$ and opens $x$'s probabilities (i.e. the output of $\mathcal{O}pen(x, \text{key}, d, \text{aux})$). We say that $Quantile$'s output is valid if $\mu \in [(\Phi_Q(x) - Q(x)), \Phi_Q(x)]$ and $\mathcal{V}_{\mathcal{C}ommit}$ accepts the probabilities and the proof.*

**Remark B.3 (Sampling from $Q$)** *By construction, for an honest sender and a uniformly random $\mu \in (0, 1]$, running $Quantile$ on input $\mu$ produces random sample $x \sim Q$ and the output is always valid. We sometimes refer to this as the distribution commitment's sampling procedure.*

*For security, the binding property of the commitment scheme means that a cheating sender can either fail the $\mathcal{Q}uantile$ procedure, or produce the appropriate element by $\widetilde{Q}$. In particular, consider a receiver who runs the sampling procedure, immediately rejects invalid outputs and runs some additional tests on $x$'s where the output was valid. The success probability of the cheating sender in passing whatever tests the receiver runs can be upper bounded (up to negligible terms) by the success probability of a sender who always produces outputs by $\widetilde{Q}$.*

**Theorem B.4 (Distribution-commitment from CRH)** *Assuming the existence of a CRH family (see Definition A.9) there exists a distribution-commitment scheme as per Definition B.1.*

*The runtime for producing the digest is $N \cdot \mathsf{poly}(\kappa, \log(N))$. The runtime for the $\mathcal{O}pen$ functionality is $\mathsf{poly}(\kappa, \log(N))$ in the RAM model.*

See the construction overview in Section 2.1. The details of the proof for Theorem B.4 follow.

**Committing.** The receiver chooses a hash function $h$ from a CRH family (as per Definition A.9), and the sender commits to the distribution $Q$ over $[N]$ by constructing a full binary tree with $N$ leaves (w.l.o.g. we assume $N$ is a power of 2, otherwise we use padding). Each node $v$ in the tree has a probability $p_v$ and a label $\ell_v$ as follows.

1. for $i \in [N]$, taking $v$ to be the $i$-th leaf in the tree, its probability and its label equal $Q[i]$.
2. for each internal node $v$ with children $u$ and $w$, its probability $p_v$ is defined to be the sum of the probabilities of its children. Its label includes its probability and a hash of the labels of its children, i.e. $\ell_v = (p_v, h(\ell_u, \ell_w))$.

   Note that the probability of each internal node is the sum of probabilities of leaves in its sub-tree, and thus the probability of the root should be 1.

The digest $d$ is the label of the root of the binary tree. The receiver checks that its probability is 1 (otherwise it rejects immediately). The auxiliary information (to be used in opening) is the entire binary tree (the probabilities and labels of all nodes).

**Opening and verifying.** To open the probability of a leaf $i \in [N]$, let $S_v$ be the set of all nodes on the path from the root to the $i$-th leaf and their siblings. The revealed pdf probability is the probability of the $i$-th leaf itself. The cdf is the sum of the $i$-th leaf's probability, and the probabilities of all internal nodes in $S_v$ that are left-children of their parent. The proof $\pi$ includes the labels of all nodes in $S_v$.

Verification proceeds as follows: for each internal node $v$ in $S_v$ with revealed label $(p_v, y)$ and children $u$ and $w$, the verifier checks that $p_v = (p_u + p_w)$ and that $y = h(\ell_u, \ell_w)$. It also checks that the revealed pdf is the probability of the $i$-th leaf and that the revealed cdf is the sum of its probabilities and the probabilities of nodes in $S_v$ that are left-children

**The extractor.** The extractor runs the given adversary $\mathcal{A}dv_2$ $\mathsf{poly}((1/\eta(\kappa)), N)$ times. It sees a polynomial collection of opened elements $x_i \in [N]$ (across all the runs), and in each opening it sees labels for internal nodes of the tree. Let $V$ be the collection of nodes (internal nodes and leaves) whose labels passed the consistency test (at least once over all the executions of $\mathcal{A}dv$): namely, all nodes on the path from them to the root (and their siblings) passed the probability-consistency test and the label-consistency test. If there are nodes in $V$ whose labels were opened differently in different executions, then the extractor fails (say it outputs a canonical distribution $\widetilde{Q}$). Looking ahead, this shouldn't happen because it means that $\mathcal{A}dv_2$ found a hash collision. Otherwise, the extractor assigns to each node in $V$ the (unique) probability that was opened for it. Observe that since these probabilities are all unique (across executions of $\mathcal{A}dv_2$) and they pass the probability-consistency test, they are consistent with a global distribution $\widetilde{Q}$ as follows:

- for each leaf $v \in V$ corresponding to the element $i \in [N]$, $\widetilde{Q}[i]$ is set to the (unique) probability that was opened for $v$.
- for each internal node in $V$ whose children (and thus also its entire subtree) are both not in $V$, let $p_v$ be the (unique) probability that was assigned to $v$. Let $L_v$ be the set of leaves in $v$'s subtree. Then for each leaf in $L_v$, corresponding to an element $i \in [N]$, $\widetilde{Q}[i] = p_v/|L_v|$.

By construction, the extractor's output $\widetilde{Q}$ is a distribution over $[N]$ (the probabilities sum to 1).

**Proof of binding.** The collision-resistance of $h$ guarantees that in all the runs of $\mathcal{A}dv_2$, including the runs of the extractor and the subsequent "critical" run, there is no node whose label is opened in two different ways (except with negligible probability). Thus, so long as the openings in the critical run are only for nodes for which the extractor also saw a successful opening, then the distribution $\widetilde{Q}$ is well defined and consistent with all of $\mathcal{A}dv_2$'s answers. The only freedom the adversary has is refusing to open certain nodes in the extractor's oracle calls, but then successfully opening them in the "critical" run in the security game. Of course, the adversary operates identically in the extractor's runs and in the critical run. Fixing the output of $\mathcal{A}dv_1$ (and conditioning on no collisions), for any node $v$ where there is at least a $(\eta/\mathsf{poly}(N))$ probability that $\mathcal{A}dv_2$ opens it correctly, the extractor will observe a correct opening w.h.p. over its $\mathsf{poly}(1/\eta, N)$ runs. Taking a union bound over the adversary's $m \leq N$ openings in its critical run, the probability that it opens a node that wasn't seen by the extractor is smaller than $\eta$. Security follows.

## C  INTERACTIVE ARGUMENT FOR DISTRIBUTION PROPERTIES

### C.1  THE VERIFIED DISTRIBUTION ORACLE PROTOCOL

**Proposition C.1** *[Verified distribution oracle protocol] Assuming the existence of a CRH family (see Definition A.9), given domain size parameter $N \in \mathbb{N}$, distance parameter $\epsilon \in (0, 1)$, black-box sample access to a distribution $D$ over domain $[N]$, and a randomized $t(N)$-time Turing Machine with oracle sample access to $D$, called* query generator machine $G^D(N, \epsilon)$*, there exists a protocol between an honest verifier and a prover such that the following holds:*

- ***Completeness.** If the prover is honest, then at the end of the interaction $q_x = Q(x)$ for all $x \in S_A = G^D(N, \epsilon)$, and the verifier accepts with all but negligible probability.*

- ***Soundness.** For every* $\mathsf{poly}$*-time prover strategy $P^*$, with all but negligible probability: either the verifier rejects; or there exists some distribution $\widetilde{Q}$ over $[N]$ such that $\delta_{TV}(D, \widetilde{Q}) \leq \epsilon$ and $q_x = \widetilde{Q}(x)$ for all $x \in S_A$.*

*The verifier runtime, $D$-sample complexity, and the communication complexity of the protocol are all $\widetilde{O}\left(\sqrt{N}/\epsilon^2 + \epsilon^{-4} + t(N)\right)$. Honest prover runs in time $\widetilde{O}(N)\mathsf{poly}(\epsilon^{-1})$.*

Consider the following protocols:

### Verified Distribution Oracle Protocol

**Input:**

- Domain size parameter $N \in \mathbb{N}$, security parameter $\kappa \in \mathbb{N}$, distance parameter $\epsilon \in (0, 1)$.

- Black-box sample access to a distribution $D$ over domain $[N]$, and a *query generator machine*, a randomized $t(N)$-time Turing Machine $G^D(N, \epsilon)$ with oracle sample access to $D$, that outputs $S_A \subseteq [N]$.

- The honest prover also gets as input an explicit description of a distribution $Q$ over $[N]$ such that $\delta_{\mathrm{TV}}(D, Q) \leq \epsilon$.

**Goal:** for every $x \in S_A$, obtain $q_x$ such there exists a distribution $\widetilde{Q}$ over domain $[N]$ that satisfies $\delta_{\mathrm{TV}}(D, \widetilde{Q}) \leq \epsilon$, and $q_x = \widetilde{Q}(x)$.
**Notations:** Let $\mathcal{G}en, \mathcal{D}ig, \mathcal{O}pen, \mathcal{V}_{\mathcal{C}ommit}$ be parts of a commitment scheme, as defined in Definition B.1.

1. **Establishing verified query access to a distribution $\widetilde{Q}$, close to $D$:**
   (a) V: generate $\mathrm{key} = \mathcal{G}en\left(1^\kappa, N\right)$, and send $\mathrm{key}$ to the prover.

   (b) P: run $\mathcal{D}ig(\text{key}, Q)$, and obtain digest $d$ and auxiliary information $\text{aux}$. Send $d$ to the verifier.

   (c) V-P: the verifier draws $O(\epsilon^{-4})$ samples according to the process described Remark B.3. For every $x$ drawn by this process, the prover provides $p_x = \Phi(x)$, $q_x = Q(x)$ and proof $\pi_x$. The verifier rejects unless $\mathcal{V}_{Commit}(x, p_x, q_x, \text{key}, d, \pi_x)$ accepts for all the samples.

   (d) V-P: interactively run the identity tester from Corollary D.4 over $d$ in the following sense: use the samples and their probability acquired in the previous step; and whenever the tester queries $Q[x]$, the verifier sends $x$ to the prover, that replies with $q_x = Q[x]$ and $p_x = \Phi(x)$, obtained from $\mathcal{O}pen(x, \text{key}, d, \text{aux})$. The verifier runs $\mathcal{V}_{Commit}(x, p_x, q_x, \text{key}, d, \pi_x)$, and rejects if the algorithm rejects. Reject if the tester rejected.

2. **Querying $\widetilde{Q}$:**

   (a) V: $S_A = G^D(N, \epsilon)$, and send it to the prover.

   (b) P: for every $x \in S_A$, run $\mathcal{O}pen(x, \text{key}, d, \text{aux})$, and send $p_x, q_x$ and $\pi_x$ to V.

   (c) V: If for all $x \in S_A$, $\mathcal{V}_{Commit}(x, p_x, q_x, \text{key}, d, \pi_x)$ didn't reject, the verifier accepts and outputs $\{(x, q_x) : x \in S_A\}$.

We argue that the protocol above satisfies the conditions of Proposition C.1, and then continue to show how this protocol can be levaraged to verify label-invariant distribution properties and languages that admit an IAP.

***Proof: the protocol is complete.*** *By Theorem B.4, the commitment scheme used in Protocol C.1 satisfies the conditions of Definition B.1. Thus, assuming the prover is honest, with probability at least $1 - \frac{1}{\eta(\kappa)}$ for every (polytime constructible) polynomial $\eta : \mathbb{N} \to \mathbb{N}$, for all $x$ queried be the verifier it holds that $q_x = Q(x)$. Since by assumption $\delta_{TV}(Q, D) \leq \epsilon$, with all by negligible probability the tester accepts.*

***Proof: the protocol is Sound.*** *By Theorem B.4, the commitment scheme used in Protocol C.1 satisfies the condition that with all but probability $\frac{1}{\eta(\kappa)}$ for any (polytime constructible) polynomial $\eta : \mathbb{N} \to \mathbb{N}$, it holds that either the verifier rejects, or there exists a distribution $\widetilde{Q}$ over domain $[N]$ such that all the verifier queries in Protocol C.1 satisfy $q_x = \widetilde{Q}(x)$.*

*By Corollary D.4, if $\delta_{TV}(D, \widetilde{Q}) > \epsilon$, the tester that the verifier ran in Step (4) of Protocol C.1 fails with all but negligible probability. And so, with overwhelming probability, either the verifier rejects, or outputs $q_x$ for all $x \in S_A$ such that $q_x = \widetilde{Q}(x)$ for some distribution $\widetilde{Q}$ over domain $[N]$ that satisfies $\delta_{TV}(D, \widetilde{Q}) \leq \epsilon$. Which concludes the soundness proof.*

The following sections show how to leverage this protocol to verify the family of $\text{polylog}(N)$ $\tau$-approximately decidable label-invariant distribution properties (see Definition A.6), and $\rho(N)$-approximately decided properties (see Definition A.8).

### C.1.1    VERIFICATION OF GENERAL DISTRIBUTION PROPERTIES THROUGH IAPS

For a distribution $Q$ over domain $[N]$, we formally describe the *representation* of $Q$ as follows:

**Construction C.2** *[String representation of Q] Fix parameters $N \in \mathbb{N}$, granularity parameter $\eta \in \Theta\left(\frac{1}{N^2}\right)$, and a distribution $Q$ over domain $[N]$ that is $\eta$-granular. Let $C : [N] \to \{0, 1\}^m$ be an error correcting code of constant rate and relative distance $\frac{1}{10}$. Define the representation $X_Q^{\eta, C} \in \{0, 1\}^{m \cdot (N/\eta)}$ as follows: for every $x \in [N]$, denote by set $t_x = \frac{Q(x)}{\eta}$, and define $X_{Q,x}^{\eta, C} \in \{0, 1\}^{m \cdot t_x}$ the string defined by concatenating $t_x$ the string $C(x) \in \{0, 1\}^m$. Set:*

$$X_Q^{\eta, C} = X_{Q,1}^{\eta, C} \circ X_{Q,2}^{\eta, C} \circ X_{Q,3}^{\eta, C} \circ \cdots \circ X_{Q,N}^{\eta, C}$$

We claim that this representation satisfies the following properties:

**Claim C.3 (Properties of Construction C.2)** *Let $Q$ be a $\eta$-granular distribution over $[N]$, and $X_Q^{\eta,C}$ be its representation as defined in Construction C.2 for $\eta = \Theta(N^{-2})$, and a constant-rate error-correcting code $C$ with relative distance at least $0.1$. The following hold:*

- *Each query to $X_Q^{\eta,C}$ can be simulated by one query to $Q$'s quantile function and one query to $Q$'s cdf.*

- *For every distance parameter $\epsilon \in (0,1)$ and distribution $Q'$ over domain $[N]$ with representation $X_{Q'}^{\eta,C}$, if $\delta_{TV}(Q,Q') > \epsilon$, then $Ham\left(X_Q^{\eta,C}, X_{Q'}^{\eta,C}\right) > \epsilon/10$*

*Proof.* Fix query $i \in [m/\eta]$. Let $Quantile_Q : [0,1] \to [N]$ be the function that for every $p \in [0,1]$ outputs the smallest $x \in [N]$ such that the *cdf* of $Q$, $\Phi_Q(x) = Q(\{y \in [N] : y \le x\})$ satisfies $\Phi(x) \ge p$. Denote $q_i = Quantile(i \cdot \eta)$. By construction, this implies that $i$ is somewhere inside the string $X_{Q,q_i}^{\eta,C}$, as defined in Construction C.2. Next, we need to know where $i$ falls inside a copy of $C(q_i)$. Again, by construction, this location is $i_{inner} = (i - \Phi(q_i - 1)/\eta) \mod m$. Finally, $X_Q^{\eta,C}(i) = C(q_i)_{i_{inner}}$. Observe that in order to output the $i$'th query, we required one query to $Quantile_Q$ and one query to the *cdf* of $Q$. This concludes te first item in the claim.

Moving to the second item, let $Q'$ be as described in the claim statement. Consider the strings $X_Q^{\eta,C}, X_{Q'}^{\eta,C} \in \{0,1\}^{m/\eta}$ to be naturally associated with strings in the space $\{0,1\}^{[1/\eta] \times [m]}$, we abuse the notation and for every $i_{outer} \in [1/\eta], i_{inner} \in [m]$, write $X_Q^{\eta,C}[i_{outer}, i_{inner}] = X_Q^{\eta,C}[(i_{outer} \cdot \eta^{-1}) + i_{inner}]$, and define also $X_Q^{\eta,C}[i_{outer}, -] \in \{0,1\}^m$ to be defined the substring of length $m$ starting at location $(i_{outer} \cdot \eta^{-1})$.

Since $Q$ and $Q'$ are of distance $\epsilon$, it must be that at least $\epsilon$-fraction $j \in [1/\eta]$ satisfy:

$$X_Q^{\eta,C}[j,-] \neq X_{Q'}^{\eta,C}[j,-]$$

Since otherwise, it means that the distributions agree on all units of mass, besides a fraction smaller than $\epsilon$, which implies that the distributions are $\epsilon$-close in $\delta_{TV}$-distance. Since $X_Q^{\eta,C}[j,-], X_{Q'}^{\eta,C}[j,-]$ are both code words in $C$, they are at constant Hamming at least $\frac{1}{10}$. We thus conclude that $Ham\left(_Q^{\eta,C}, X_{Q'}^{\eta,C}\right) \ge \epsilon/10$. $\square$

**Definition C.4 (The $\delta_c$-tolerant language of distribution property $\Pi$)** *Let $\Pi$ be some distribution property, and let $\eta(N) = \Theta(N^{-2})$ be a granularity function. Fix a constant rate and distance $\frac{1}{10}$ ECC family, such that $C_N : [N] \to [m(N)]$. We define the $\delta_c$-tolerant language of $\Pi$ to be $\mathcal{L}_\Pi = (\mathcal{L}_{\Pi,N})_N$ such that:*

$$\mathcal{L}_{\Pi,N}^{\delta_c} = \left\{ X_Q^{\eta(N),C_N} : \delta_{TV}(Q, \Pi_N) \le \delta_c \right\}$$

**Claim C.5 (Representation Decision Algorithm)** *For every $N \in \mathbb{N}$, let $m(N)$ and $\eta(N)$ be as defined in Construction C.2. Every distribution property $\Pi$ that is $\rho(N)$-approximately decided in $\mathsf{poly}(N)$ time admits a $\mathsf{poly}$-time algorithm $\mathcal{T}est_\Pi$ that takes as input $N$, a string $X \in \{0,1\}^{m/\eta}$, and distance parameter $\delta_c$, such that the following hold:*

- *If there exists a distribution $Q$ such that $X = X_Q^{\eta(N),C_N}$, and $\delta_{TV}(Q, \Pi_N) \le \delta_c$, then $\mathcal{T}est_\Pi$ accepts.*

- *If $X$ doesn't encode a distribution over domain $[N]$, or there exists a distribution $Q$ such that $X = X_Q^{\eta(N),C_N}$, but $\delta_{TV}(Q, \Pi_N) \ge \delta_c + \rho(N)$, then $\mathcal{T}est_\Pi$ rejects.*

*Proof.* We define the algorithm:

- $\mathcal{T}est_\Pi$ checks that $X \in \{0,1\}^{m/\eta}$ encodes a valid distribution $Q$, and rejects otherwise. This is done by considering $X$ as a string over the space $\{0,1\}^{[1/\eta] \times [m]}$, and that for every $i \in [1/\eta]$, $X(i,-)$ is a codeword in $C$. Denote the decoding of $X(i,-)$ as $x_i$. It then checks that for all $i$, $x_i \le x_{i+1}$, and rejects otherwise.

- Let $\mathcal{D}ist_\Pi$ be as defined in Definition A.8. If the first test passed, $\mathcal{T}est_\Pi$ reconstructs the distribution $Q$ that satisfies $X = X_Q^{\eta(N),C_N}$ from the decoded string, and runs the $\mathcal{D}ist_\Pi$ over $Q$ with accuracy parameter $\rho(N)$. If the output $\delta$ of $\mathcal{D}ist_\Pi$ satisfies $\delta \leq \delta_c + \rho(N)$, it accepts, and otherwise, it rejects.

This concludes the proof. □

This immediately implies the following:

**Corollary C.6** *For every distribution property $\Pi$ that is $\rho(N)$-approximately decided in $\mathsf{poly}(N)$, for any distance parameters $\delta_c \in (0,1)$, the promise problem of being in $\mathcal{L}_{\Pi,N}^{\delta_c}$, or $\rho(N)$-far from it, is in $\mathbb{P}$.*

We thus provide the protocol behind Theorem 1.1. Given $\rho$-approximately decided property $\Pi$, and distance parameters $\delta_c, \delta_f \in (0,1)$. Run Protocol C.1 with parameters $N$, $\epsilon = \frac{\delta_f - \delta_c}{10} > \rho(N)$, and:

1. The prover sets $Q = D$.
2. After running Step $(1)$ of the protocol, run the communication phase of the IAP described in Theorem A.11 with distance parameter $\left(\frac{\delta_f - \delta_c}{20}\right)$, and with respect to the language $\mathcal{L}_{\Pi,N}^{\delta_c}$, and the string $X_C^{\eta,Q}$. Obtain the queries to the string $X_C^{\eta,Q}$ through the IAP.
3. Set the query set for $Q$ to be the elements in $Q$ relevant to the locations queried by the protocol by the IAP in the representation $X_C^{\eta,Q}$, as explained in Claim C.3 (recall that that each such query to the representation can be simulated by queries provided by the *distribution commitment scheme*). Obtain verified openings from the prover,
4. Check that the IAP predicate accepts the queries, and reject otherwise.

We argue that this protocol satisfies the conditions of Theorem 1.1:

- Assume $\delta_{\mathrm{TV}}(D, \Pi_N) \leq \delta_c$. Since and $D = Q$, and the prover is honest, the first stage of Protocol C.1 doesn't result in rejection with overwhelming probability. Then, Since $\delta_{\mathrm{TV}}(D, \Pi_N) \leq \delta_c$, and the queries to the representation string are correct according to $D$, from the completeness of the IAP, the verifier accepts with high probability.

- Assume $\delta_{\mathrm{TV}}(D, \Pi_N) \geq \delta_f$, then, assuming that Step $(1)$ of Protocol C.1 passed, then with overwhelming probability there exists a distribution $\widetilde{Q}$ such that $\delta_{\mathrm{TV}}(\widetilde{Q}, D) \leq \frac{\delta_f - \delta_c}{10}$ and all the prover's openings are according to $\widetilde{Q}$. Then, since $\delta_{\mathrm{TV}}(D, \Pi_N) \geq \delta_f$, by the triangle inequality, it holds that $\delta_{\mathrm{TV}}(\widetilde{Q}, \Pi_N) \geq \delta_f - \frac{\delta_f - \delta_c}{10}$. By Claim C.3, this implies that $\mathrm{Ham}\left(X_C^{\eta,\widetilde{Q}}, X_C^{\eta,Q}\right) > \frac{1}{10} \cdot \frac{9(\delta_f - \delta_c)}{10} > \frac{\delta_f - \delta_c}{20}$ for any $Q$ that is $\delta_c$ close to $\Pi_N$, and so, by the soundness condition of the IAP, the predicate associated with it rejects with high probability.

## D  THE IDENTITY TESTER

**Theorem D.1 (Optimal Identity Tester Valiant & Valiant (2014); Goldreich (2020))** *For every fixed distribution $Q$ over $[N]$, there exists an algorithm that given $N$, oracle access to $Q$, and black box sample access to a distribution $D$ over domain $[N]$, draws $O(\sqrt{N}\epsilon^{-2})$ samples from $D$, and:*

- *If $D = Q$, the algorithm accepts with high probability.*

- *If $\delta_{TV}(D, Q) > \epsilon$, the algorithm rejects with high probability.*

We give a short review of the algorithm behind Theorem D.1, and then explain how it is adapted to the setting of our result. We follow the algorithm described in Goldreich (2020).

First, we describe the algorithm under the following assumptions, that we later remove:

- For some $m = O(N)$, assume $Q$ is $\frac{1}{m}$-*granular*, i.e. for all $x \in [N]$, $Q(x) = m_x \cdot \frac{1}{m}$, where $m_x \in \{0, 1, \ldots, m\}$.

- Assume $Q$ has *full support*, $\text{Supp}(Q) = [N]$.

Consider the set $S = \{\langle x, j \rangle : x \in [N], j \in [m_x]\}$, and for every $x \in [N]$ define the random variable $F_Q(x_0)$ to be a uniformly chosen tuple from $S_{x_0} = \{\langle x_0, j \rangle : j \in [m_x]\}$.

Observe that the distribution over $S$ obtained by first drawing $x \sim Q$, and then selecting $\langle x, j \rangle$ through $F_Q(x)$ is the uniform distribution over $S$, since every $\langle x, j \rangle \in S$ is sampled with probability $Q(x) \cdot \frac{1}{|S_x|} = \frac{m_x}{m} \cdot \frac{1}{m_x} = \frac{1}{m}$. In other words, if $D$ is distributed according to $Q$, then $F_Q(D) = U_S$. In fact, this transformation preserves the *total variation distance* in the following sense: if $\delta_{\text{TV}}(D, Q) = \delta$, then $\delta_{\text{TV}}(F_Q(D), U_S) = \delta$, since:

$$\delta_{\text{TV}}(F_Q(D), U_S) = \frac{1}{2} \sum_{x \in [N]} \sum_{j \in m_x} \left| F_Q(D)(\langle x, j \rangle) - \frac{1}{m} \right| \tag{1}$$

$$= \frac{1}{2} \sum_{x \in [N]} \sum_{j \in [m_x]} \left| D(x) \cdot \frac{1}{m_x} - \frac{m_x}{m} \cdot \frac{1}{m_x} \right| \tag{2}$$

$$= \frac{1}{2} \sum_{x \in [N]} \sum_{j \in [m_x]} \frac{1}{m_x} \left| D(x) - \frac{m_x}{m} \right| \tag{3}$$

$$= \frac{1}{2} \sum_{x \in [N]} \sum_{j \in [m_x]} \frac{1}{m_x} |D(x) - Q(x)| \tag{4}$$

$$= \frac{1}{2} \sum_{x \in [N]} |D(x) - Q(x)| \tag{5}$$

$$= \delta_{\text{TV}}(D, Q) \tag{6}$$

Note that we used the assumption that $Q$ has full support on the second inequality when dividing by $m_x$, and that it is $\frac{1}{m}$-*granular* on the fourth inequality to argue that $Q(x) = \frac{m_x}{m}$ for all $x$.

And so, testing that $D = Q$ is reduced to testing that $F_Q(D)$ is uniform. Testing that a samplable distribution over domain of size $m$ is uniform over the entire domain or $\epsilon$-far from it in $\delta_{\text{TV}}$-distance is a fundamental well-researched problem in distribution testing, that is known to require $\Theta(\sqrt{m}\epsilon^{-2})$ samples (see Theorem A.4). Observe that $F_Q(D)$ as a distribution over $S$ is indeed samplable by first sampling $x \sim D$, querying $Q(x)$, computing $m_x$ from it, and then sampling $j$ uniformly from $[m_x]$.

**Removing assumptions over $Q$.**    We explain how to overcome the two assumptions given above:

- **Accounting for $Q$ with partial support.** Consider the distributions $Q' = \frac{1}{2}Q + \frac{1}{2}U_{[N]}$ (the distribution that assign $x \in [N]$ the probability $\frac{1}{2}Q(x) + \frac{1}{2N}$), and $D' = \frac{1}{2}D + \frac{1}{2}U_{[N]}$. These two distributions are of *full support*, and satisfy $\delta_{\text{TV}}(D', Q') = \frac{1}{2}\delta_{\text{TV}}(D, Q)$. Moreover, $D'$ is samplable given that $D$ is samplable, and $Q'$ can be queried given oracle access to $Q$. Thus, assuming that $Q'$ is $\frac{1}{m}$-*granular* for some $m = O(N)$, in order to test whether $D = Q$ or $\delta_{\text{TV}}(D, Q) > \epsilon$, suffice to test that $D' = Q'$ or $\delta_{\text{TV}}(D', Q') > \epsilon/2$, by sampling $F_{Q'}(D')$ and running the uniformity test as explained above, incurring only constant overhead in the sample complexity.

- **Accounting for non-granular $Q$.** Assume that $Q'$ as described above isn't $\frac{1}{m}$-*granular* for $m = O(N)$. Set $m = 6N$, and for every $x \in [N]$ define $\theta_x = \frac{\lfloor Q'(x)/(1/m) \rfloor \cdot (1/m)}{Q'(x)}$. We now define the distributions $D''$ and $Q''$ over $[N+1]$:

    - For every $x \in [N]$, $Q''(x) = \theta_x \cdot Q'(x)$, and $Q''(N+1) = 1 - \sum_{x \in [N]} \theta_x \cdot Q'(x)$.
    - For every $x \in [N]$, $D''(x) = \theta_x \cdot D'(x)$, and $D''(x) = 1 - \sum_{x \in [N]} \theta_x \cdot D'(x)$

Note that if $D' = Q'$, then $D'' = Q''$, and if $\delta_{\mathrm{TV}}(D', Q') > \epsilon/2$, then:

$$\delta_{\mathrm{TV}}\left(D'', Q''\right) = \frac{1}{2} \sum_{x \in [N+1]} |D''(x) - Q''(x)| \tag{7}$$

$$= \frac{1}{2} \left( \sum_{x \in [N]} \theta_x \left| D'(x) - Q'(x) \right| + |D''(N+1) - Q''(N+1)| \right) \tag{8}$$

$$\geq \frac{1}{2} \left( \sum_{x \in [N]} \theta_x \left| D'(x) - Q'(x) \right| \right) \tag{9}$$

$$\geq \frac{2}{3} \cdot \frac{\epsilon}{2} \tag{10}$$

$$\geq \frac{\epsilon}{3} \tag{11}$$

Where the second to last inequality is justified by the fact that $Q'(x) \geq \frac{1}{2N}$ for all $x \in [N]$, and thus, $\theta_x \geq 1 - \frac{1/6N}{Q'(x)} \geq \frac{2}{3}$. We conclude that in order to determine that $D = Q$ or $D$ is $\epsilon$-far from $Q$, suffice to to determine if $D'' = Q''$ or $D''$ is $\epsilon/3$-far from $Q''$. Observe that given oracle access to $Q$, $\theta_x$ can be computed for every $x \in [N]$, so both oracle access to $Q''$ and sample access to $D''$ can be simulated. Lastly, assuming without loss of generality that $N$ is divisible by 6, $Q''$ is $\frac{1}{m}$-*granular* for $m = 6N$.

Let $Q''$ and $D''$ be as defined in the above proof. For every $x \in [N]$, querying $Q''(x)$ can be be done by querying $Q(x)$ and computing $Q'(x)$ and $\theta_x$ from it. However, in order to query $Q''(N+1)$, the tester needs to compute $\sum_{x \in [N]} (1 - \theta_x) Q'(x)$, which involves querying $N$ locations in $Q$, increasing the tester's runtime significantly. The original setting for this tester was concerned with matching the $D$-sample complexity of the tester with the lower bound of $\Omega\left(\sqrt{N}\epsilon^{-2}\right)$, regardless of the runtime of the tester. Since we want the runtime of the tester to also be of the same magnitude, we circumvent this obstacle by estimating $Q''(N+1)$ up to a very small error. This can be done with samples drawn according to $Q$, alongside the query access to $Q$. Concretely:

**Claim D.2** *Fix a distribution $Q$ over domain $[N]$, and let $Q''$ be as defined in the proof of Theorem D.1. There exists an algorithm with oracle access to $Q$ that upon obtaining $O(\epsilon^{-4})$ samples from $Q$, approximates $Q''(N+1)$ up to additive error of $O(\epsilon^2)$ with high probability.*

We prove Claim D.2:

Consider the following algorithm: it draws $s = O(\epsilon^{-4})$ samples according to $Q$, and for every sample $x_i$ drawn, it queries $Q(x_i)$, computes $Q'(x_i) = \frac{1}{2} Q(x_i) + \frac{1}{2N}$, and then, computes $1 - \theta_{x_i} = 1 - \frac{\lfloor Q'(x)/(\gamma/N) \rfloor \cdot (\gamma/N)}{Q'(x_i)}$. It then outputs $\sum_{i \in [s]} (1 - \theta_{x_i}) Q'(x_i)$.

Observe that $Q''(N+1) = \sum_{x \in [N]} (1 - \theta_x) Q'(x) = \mathbb{E}_{x \sim Q'} [1 - \theta_x]$. Since the random variable $(1 - \theta_{x_i})$ can only obtain values in the interval $(0, 1)$, by Hoeffding's Inequality:

$$\Pr\left( \left| \frac{1}{s} \sum_{i \in [s]} (1 - \theta_{x_i}) - Q''(N+1) \right| \geq \frac{1}{\epsilon^2} \right) \leq 2e^{-2s/\epsilon^4}$$

And so, since we took $s = O(\epsilon^{-4})$ the algorithm described above provides an approximation of $Q''(N+1)$ up to additive error $\epsilon^2$ with high probability.

**Remark D.3** *An $\epsilon^2$ additive approximation of $Q''(N+1)$ suffices for the purpose of the algorithm described in the proof of Theorem D.1, since it only effects the estimated size of the set $S$ by a multiplicative factor of $\epsilon^2$. However, this doesn't effect the uniformity tester. The reader is reffered to Cannone Canonne (2015) for further detail.*

Therefore, using samples from $Q$ to estimate $Q''(N+1)$, we conclude the following:

**Corollary D.4** *For every fixed distribution $Q$ over $[N]$, there exists an algorithm that given parameter $N$, oracle access to $Q$, as well as sample access to $Q$, and black box sample access to a distribution $D$ over domain $[N]$, draws $O(\sqrt{N}\epsilon^{-2})$ samples from $D$, $O\left(\epsilon^{-4}\right)$ samples from $Q$, and:*

- *If $D = Q$, the algorithm accepts with high probability.*

- *If $\delta_{TV}(D, Q) > \epsilon$, the algorithm rejects with high probability.*

*The runtime of the algorithm is $O\left(\epsilon^{-4} + \sqrt{N}\epsilon^{-2}\right)$*

**Remark D.5** *We only consider $\epsilon$ such that $\epsilon = \omega(N^{-1/4})$, since every smaller value provable required super linear sample complexity, and so $O(\epsilon^{-4} + \sqrt{N}\epsilon^{-2}) = O(\sqrt{N}\epsilon^{-2})$. Also note that the completeness and soundness error can be made negligibly by amplification through repetition* $\mathsf{polylog}(N)$ *times.*

