# OpenReview forum: "How to Verify Any (Reasonable) Distribution Property: Computationally Sound Argument Systems for Distributions"
_ICLR.cc/2025/Conference — ICLR 2025 Poster_

### Official Review · Reviewer_2JWS · 2024-11-06

**Soundness:** 3
**Presentation:** 3
**Contribution:** 2
**Rating:** 6
**Confidence:** 4

**Summary:**

The paper describes how a verifier can check that a distribution which they can sample from has (up to small error) a certain (polynomial time verifiable) property in O(sqrt(n)polylog(n)) time with support from an untrusted prover that already knows the distribution. The protocol consists of the prover commiting to (an approximation of) the true distribution, then proving to the verifier that this matches what they are sampling in a way that only needs O(sqrt(n)) samples, finally the prover proves that the committed distribution has (or rather is clsoe in TV distance to having) the desired property.

**Strengths:**

The problem is natural and the solution is clean and at least in some senses optimal.

The presentation does a good job of clearly explaining what it does ( althoguh a lot of details are left out of the main part).

**Weaknesses:**

I would like to have mroe details on the identity testing part, I'm guessing that this is not particularly novel to this work but there should at least be a reference in the section describing it to where more information can be found (edit: I've found the reference in sec 2.0 but a reminder of it later would be good as well). It would also be good to have more explanation of how it works, e.g. to what extent do the queries to the committed distribution depend on samples? would it be possible to make that part non-interactive with the FS heuristic at the cost of assuming your CRH is an RO?

This paper is largely jsut a combination of powerful ideas from previous papers, though there is some non-trivial work in choosing the represntations for the distributions to allow the application of the previous work.

This work is unlikely to work well in practice due to the constants being very big in some of the used tools (e.e. PCP theorem).

**Questions:**

See the questions embedded in the weaknesses.

---

> ### Author Response · Authors · 2024-11-24
>
> Thank you for your insightful feedback. Following your comments, we added more details regarding the commitment scheme, as well as added an entire appendix dedicated to the identity tester (see Appendix D).
>
> We do not agree that our work is just a combination of powerful ideas from previous papers: of course, powerful prior results play important roles (as they do in most research papers), but for example the idea of a distribution-commitment is a novel contribution and we believe it can have applications in future work.
>
> Regarding the practicality of our work, we remark that there is a growing literature buildinging on theoretical proof-system constructions (some of them using PCP theorems) to construct efficient systems. In the context of block-chains, such systems are running at internet scale. We are optimistic that future work can leverage our results and tools from these literatures to obtain more efficient and practical protocols for our setting.
>
> Regarding your questions about identity testing: the main role of the samples is to verify that the committed probabilities are consistent with the actual input distribution, which is only accessible through sample access. Once this is verified, the verifier is guaranteed that the query requests it makes to the committed distribution are indeed correct (or at least close enough to be correct). Therefore, the queries aren’t directly related to the samples, but the samples are used to verify that the query answers are indeed consistent with the correct distribution.
> In order to utilise the FS heuristic we would need the protocol to be public coin. Currently this isn’t the case, since the verifier doesn’t simply send random bits, but samples from a (not-necessarily uniform) distribution. The question of whether our protocol can be made public-coins is very interesting (especially in light of recent work on the topic [Herman24]), we added a comment about this in the paper, see line 178.

---

### Official Review · Reviewer_BGYN · 2024-11-07

**Soundness:** 4
**Presentation:** 4
**Contribution:** 3
**Rating:** 8
**Confidence:** 3

**Summary:**

A recurring theme in theoretical computer science is that *verifying* an answer is often (computationally) easier than deriving the answer *from scratch*. This work illustrates this principle in the context of distribution testing. Let $\mathcal{P}$ be a distribution property (i.e., set of distributions) such that with full knowledge of a given distribution $Q$, I can efficiently decide whether $Q \in \mathcal{P}$ or not. The paper’s main result shows that If someone *else* (”Prover”) has already put in extensive effort to fully identify a distribution $Q$ and I (”Verifier”) only have sample-access to $Q$, then I can test whether $Q$ satisfies the property $\mathcal{P}$ with far less effort by engaging in an interactive protocol with the Prover. This verification is feasible even without any trust in the Prover and is strictly more efficient than the naive solution of the Prover simply sending over the full representation of $Q$.

The lack of trust in the Prover introduces several challenges. Let $\tilde{Q}$ be the representation of the “distribution” on $[N]$ the Prover holds (quotations since the $\tilde{Q}$ may not even be a valid distribution) and let $Q$ be a distribution on $[N]$ which the Verifier has only sample-access to. Here, the support size $N$ is the index with respect to which computational complexity is defined. In the absence of trust, the Verifier must check that 1) $\tilde{Q}$ is a valid distribution and 2) $\tilde{Q} = Q$. The second part uses the identity (a.k.a. goodness-of-fit) testing algorithm by Goldreich (2017) which has $\sqrt{N}$ sample complexity. The Prover’s overall honesty in responding to Verifier’s queries is enforced through cryptographic commitment schemes, whose security is based on collision-resistant hash functions.

Once the Verifier checks that $\tilde{Q} = Q$, then it remains to check that $\tilde{Q} \in \mathcal{P}$. The deterministic decision $Q \in \mathcal{P}$ potentially requires poly($N$) time to compute. However, PCPs allow the Verifier to probabilistically check this much more efficiently, incurring only poly($\log(N)$) additional communication cost for checking $\tilde{Q} \in \mathcal{P}$ on top of the $\sqrt{N}$ needed for identity testing.

**Strengths:**

I believe this is a well-executed work with solid contributions for reasons discussed below.

**Exceptional exposition.** The paper is impeccably written, providing a highly accessible explanation of non-trivial techniques while maintaining full formality. Papers at the intersection of computational complexity and learning theory frequently overlook considerations related to input representation and finite precision. This paper addresses all such subtleties explicitly and cleanly. Moreover, the high-level ideas for the interactive protocol are easy to follow as they are presented in a modularized manner, with each module highlighting key insights.

**Novelty of ideas.** The use of cryptographic commitment schemes to prevent cheating of the Prover seems like a natural approach and may be standard in the field, although I am not familiar with the interactive proofs literature. However, combining commitment schemes with identity testing to achieve $\sqrt{N}$ sample complexity appears novel since it involves distilling the *essence* of previously known identity testing algorithms. As shown in Section 2 (line 301), the authors analyze what the identity tester *actually* needs to know about $Q$, rather than its full truth table. Furthermore, this interactive protocol works for *any efficiently decidable* distribution property which is  a substantial improvement over previous work.

**Weaknesses:**

Further discussion on the following aspects would be beneficial for the paper.

**Distribution properties that are *not* efficiently decidable.** To develop intuition for efficiently decidable distribution properties, it would be good to understand what properties are *not* efficiently decidable. Also, if we restrict to singleton sets, what is an example of a (sequence of) distributions the TV distance to which is not efficiently computable? Few candidates that come to mind are

- (Uncomputable?) A sequence $(Q_N)$ which is uniform over a subset $S_N \subseteq [N]$ where $j \in S_N$ if and only if $M_j$ halts on input $N$, where $M_j$ is the $j$-th Turing machine enumerated by a universal Turing machine.

- (Inefficient?) A class of "perfectly balanced" distributions $\mathcal{P}_N$, where $Q \in \mathcal{P}_N$ if there exists a subset $S \subset [N]$ such that $Q(S) = Q([N]\setminus S)$.

**Verifying ERMs (line 198)?** The term "poly-time" ERM algorithm is unclear. What is the polynomial with respect to? Is it the support size $N$? The notion of a "poly-time" ERM algorithm sounds weird to me since for most interesting hypothesis classes, ERM is not known to be poly time (in $n = \log N$ where we identify $[N] = \\{0,1\\}^n$). What is the "efficiently decidable property" that corresponds to this ERM verification setup? What is the role of the hypothesis class $\mathcal{H}$?

**Questions:**

1. In Theorem 1.1, it seems like the sample complexity of the Verifier is $\sqrt{N} \cdot \mathrm{poly}(\kappa)$, but since we assume $\kappa$ to be polynomially related to $N$, shouldn't sample complexity technically be $N^{1/2+c}$ for an arbitrary constant $c > 0$?

2. What properties are *not* efficiently decidable? For example, are the following properties efficiently decidable (or even decidable at all)?
    - (Uncomputable?) A sequence $(Q_N)$ which is uniform over a subset $S_N \subseteq [N]$ where $j \in S_N$ if and only if $M_j$ halts on input $N$, where $M_j$ is the $j$-th Turing machine enumerated by a universal Turing machine.
    - (Inefficient?) A class of "perfectly balanced" distributions $\mathcal{P}_N$, where $Q \in \mathcal{P}_N$ if there exists a subset $S \subset [N]$ such that $Q(S) = Q([N]\setminus S)$.

3. What is a "poly-time" ERM algorithm? What is the polynomial with respect to?

4. What is the "efficiently decidable property" that corresponds to the ERM verification setup (line 198)? What is the role of the hypothesis class $\mathcal{H}$?

---

> ### Author Response · Authors · 2024-11-24
>
> Thank you for your insightful feedback. We are gratified by your positive opinion about the results and the writing. As to your questions:
> You are correct. We will add a remark rephrasing the sample complexity as you suggested, on top of the current phrasing. We note here that the current phrasing is still relevant, as it is possible to also assume stronger CRH’s, where \kappa is poly-logarithmic related to N (but we focus on polynomial security parameters throughout the work).
> The examples given are indeed not efficiently decidable (and the first is uncomputable). We added such examples to the writeup (see footnote 2 on page 2).
>
> When we say “poly-time” ERM algorithm we mean to say that the algorithm is poly-time with respect to the sample size (the training set). Note that the hypothesis class H dictates both the sample size as well as the specific ERM algorithm to be used.
> When we apply our protocol in this setting, the distribution we consider is the distribution obtained by sampling a random element from the training set (so the size of the domain is the size of the training set, and the verifier’s runtime is sublinear in the size of the training set).
>
> Let D be a uniform distribution over a set of samples used for training. For poly-time ERM algorithm with respect to hypothesis class H, consider the following efficient distribution property: “the best loss of a predictor in H over the distribution D is roughly $\epsilon$ ”. This is efficiently decidable since we could run the (efficient) ERM algorithm on the description of distribution D  (capturing the entire training set), obtain a predictor with roughly optimal loss, and verify that the loss is indeed close to $\epsilon$. If the prover provided a predictor instead of claim about the loss of the optimal predictor, we could just calculate the loss of the predictor first, then proceed as above.

---

### Official Review · Reviewer_ZJhC · 2024-11-07

**Soundness:** 3
**Presentation:** 2
**Contribution:** 3
**Rating:** 8
**Confidence:** 3

**Summary:**

This paper considers the problem of verifying whether an unknown discrete distribution satisfies certain properties with a prover. The authors show that it is possible to verify a wide class of properties using a sublinear number of samples and running time. Without a prover, even testing some properties requires quasi-linear samples, which is a big improvement.

**Strengths:**

1. The problem of verifying a claim from another party is theoretically important and has practical applications.
2. The authors achieve sublinear time, communication, and sample complexity for the verifier. In contrast, (tolerant) testing without a prover often requires quasi-linear samples. This is a substantial improvement over previous works. It is also an interesting theoretical result that shows verifying with a prover can be easier than doing testing without a prover.
3. The problem formulation and results are very general and can be applied to many testing and learning problems.

**Weaknesses:**

1. Is the $\tilde{O}(N)$ sample complexity for an honest prover also optimal?
2. I prefer to have some key technical results and lemmas when presenting the technical steps.
3. Theorem B.4. does not have full proof. What is its role in the result?

**Questions:**

Seet the first point in weaknesses.

---

> ### Author Response · Authors · 2024-11-24
>
> Thank you for your insightful feedback. We are gratified by your enthusiasm about the results.
>
> The sample complexity of the honest prover is quasi-linear in N and polynomial in the inverse of the distance parameter $\varepsilon$. The sample complexity of the prover cannot be smaller than the complexity of testing (since running the prover and the verifier together gives a tester), we added a comment about this at line 140. Many natural distribution properties require $\Omega(N/ \log N)*poly(1/\varepsilon)$ samples to test, e.g.: the property of being at distance \delta from uniform. Therefore, the sample complexity of our honest prover strategy is optimal up to $poly(\log N, 1/\epsilon)$ multiplicative factors.
>
> Following your remarks, we added more technical details, elaborating on Theorem B.4. as well as the identity tester (expanded on Appendix B, and added C and D). We hope that these added details make the work clearer, and we will be happy to hear from you if there are still details missing in your opinion.

---

> > ### Comment · Reviewer_ZJhC · 2024-11-26
> >
> > Thanks for your response. My evaluation remains the same.

---

### Official Review · Reviewer_zrGv · 2024-11-08

**Soundness:** 3
**Presentation:** 3
**Contribution:** 2
**Rating:** 6
**Confidence:** 4

**Summary:**

The paper introduces an interactive protocol for the verification of distribution properties, computationally sound. The protocol is performed between a verifier and an untrusted prover, where both have sampling access to some unknown distribution. The major contribution of this work is an argument system that can be used in verifying any property such that, given the full description of a distribution, one may approximately compute its distance from the property in polynomial time.The protocol has a communication complexity and verifier runtime of roughly  O( N​/eps^2 ) which is nearly optimal. This yields a quadratic speedup over prior approaches with quasi-linear sample complexity. The authors also provide applications to testing properties like monotonicity and juntas and extend their approach to properties in NP.

**Strengths:**

Quadratic Speedup: The protocol provides a quadratic improvement over existing works, closing the sample complexity and runtime of the verifier to near optimality.

Broad Applicability: It can verify a wide range of properties, including those that are not label-invariant, expanding the scope beyond previous work.

Detailed Explanation: The paper offers clear explanations and technical overviews, making complex concepts accessible.

Extensions and Applications: The authors illustrate how the protocol can be applied to many different properties and extend it to properties in NP, showing its versatility

**Weaknesses:**

Cryptographic Assumptions: The reliance on collision-resistant hash functions means that the soundness of the protocol is predicated upon some cryptographic assumptions, which may not be valid in every setting.

Limited Practical Evaluation: The paper does not include any empirical results or practical demonstrations showing the protocol's performance on real data.

Omitted Technical Details: The book omits some proofs and detailed constructions, like the full proof of Theorem B.4, and such an omission could impair complete understanding of the implementation.

**Questions:**

Can the authors elaborate more on how the distribution commitment scheme actually prevents a cheating prover from misleading the verifier?

How will the choice of collision-resistant hash function eVect the performance and security of the protocol?

Are there scalability issues when this protocol is applied to distributions with very large support sizes, and how might they be addressed?

---

> ### Author Response · Authors · 2024-11-24
>
> Thank you for your insightful feedback. We hope this work can contribute to construction of practical tools for ensuring and promoting trust in machine learning systems. Following your comments, we elaborated on Theorem B.4. and added proof and more details following it.
>
> As to your questions:
>
> Any prover that wants to mislead the verifier has to lie about the probabilities of elements sampled by the verifier. This can be done by either one of the following two methods: (1) committing to an incorrect distribution before seeing the verifier’s samples, then opening commitments to these incorrect values upon the verifier’s request - this will be caught by the “identity tester” the verifier employs; (2) commit to some values (correct or otherwise) for the probabilities of each element, but then adaptively “opening” the commitment to values of the prover’s choice, after seeing the verifier’s samples. The distribution commitment scheme prevents this type of cheating. Elaboration on the nature of the commitment scheme can be found in the section that we just added following Theorem B.4..
>
> The collision-resistant hash function is employed by both the prover and the verifier in our protocol. A CRH that can be computed very quickly will speed up both parties (and vice versa). The choice of security parameter also affects efficiency as described in the result statements. If a hash function is believed to be very secure, potentially one can choose a smaller security parameter and obtain improved efficiency.
>
> The sample complexity and runtime of the verifier depend on the support size of the distribution. If the distribution is of large support, the complexity of the verifier will inevitably increase accordingly (by a square-root factor). This is inherent: as noted in the paper, there is a matching lower bound on the verifier’s sample complexity. Our work provides a general tool for many distribution properties, yet we believe that future works might explore how certain assumptions over the distribution, or the learning task at hand might yield a better tradeoff between the verifier complexity and the support size of the distribution.

---

> > ### Comment · Reviewer_zrGv · 2024-11-26
> >
> > Thanks for your response! My rating remains the same. Please make sure to include your answers above in the final version if accepted.

---

### Official Review · Reviewer_K79e · 2024-11-08

**Soundness:** 4
**Presentation:** 4
**Contribution:** 4
**Rating:** 8
**Confidence:** 3

**Summary:**

This paper considers a setting where an untrusted prover and verifier both have sample access to an unknown distribution, and the prover wishes to convince the verifier that the distribution has some property. The prover does so using an interactive protocol, which is efficient in the sense that the communication complexity and sample complexity of the verifier are low. In particular, their protocol applies to some properties where testing (from scratch, without a prover) provably requires a higher sample complexity than that of the verifier.
	Their protocol has two components. The first is a distribution-commitment, a new cryptographic primitive that allows a prover to succinctly commit to a distribution and verifiably answer natural queries, such as the probability of any element or the cdf. The second component is a reduction to interactive arguments of proximity, using this distribution commitment. That is, they show a protocol where the prover commits to the distribution. The verifier then tests the correctness of this commitment by sampling from the unknown distribution. The prover and verifier then engage in an interactive argument of proximity to test that the committed distribution indeed has the claimed property.

**Strengths:**

The paper shows an original and strong result, that a large class of distributional properties can be proven efficiently. It does so using interesting techniques; in particular, it introduces the distribution commitment which appears to be a useful primitive for future protocols. The distribution commitment allows them to leverage existing work on IAPs to construct their protocol. The ideas are elegant and clearly presented.

**Weaknesses:**

While the high-level ideas are presented clearly, the details are a bit lost due to the submission format. For example, there is no proof of their distribution oracle construction. It would be helpful to explain how their commitment ensures that the prover doesn’t commit to conflicting probabilities and cdf values.

Distribution commitments seem even closer to vector commitments than cryptographic accumulators. It would be nice to see this discussed in 2.1.

**Questions:**

How does one ensure that the committed cdf values are consistent with the committed probabilities?

How do distribution commitments relate to random variable commitments ([BGKW24])? Can they be used to construct random variable commitments?

[BGKW24]: https://eprint.iacr.org/2024/938.pdf

---

> ### Author Response · Authors · 2024-11-24
>
> Thank you for your insightful feedback. We are gratified that you found the results to be original and strong. We added an elaboration on Theorem B.4. with a proof and construction, explaining the commitment scheme more carefully and explicitly, as well as how the verifier obtains access to the cdf. In broad strokes, we argue that assuming collision-resistant hashing, any digest the prover provides either is rejected by the verifier with high probability or represents some specific distribution $\tilde{Q}$ (this is formalised through the existence of the extractor, see line 905), that the verifier can verifiably access (see line 998 for more information, and specifically about obtaining the cdf).
>
> We want to thank the reviewer for bringing BGKW24 to our attention. We added a remark in the paper comparing the results in Section 1.4 under “further related works”. The works are quite different: BGKW24 consider a setting where both parties have full information about some distribution (e.g. over noise values), and the verifier needs to be convinced that a value (hidden inside a commitment) was indeed drawn from the known distribution. In contrast, in our work, the verifier doesn’t know the distribution but can only sample from it. Our distribution-commitments allow an untrusted prover to commit to the entire description of the unknown distribution (rather than a single unknown sample from a known distribution). It is not clear if or how our distribution commitments can be used to construct RV commitment schemes.

---

### Official Review · Reviewer_JVRG · 2024-11-09

**Soundness:** 3
**Presentation:** 3
**Contribution:** 2
**Rating:** 6
**Confidence:** 3

**Summary:**

This paper gives efficient interactive verification protocols for distribution property testing. Specifically, the goal is to design a communication protocol between a prover and a verifier, such that the verifier, given sample access to an unknown distribution $D$ and the ability to make certain queries to the prover, is able to verify the membership of $D$ in a class/property of probability distributions. A tolerant verifier parametrized by $\varepsilon_c$ and $\varepsilon_f$ is guaranteed to verify membership of distributions that are at most  $\varepsilon_c$-close in total variation distance to some distribution in the class and to reject  $\varepsilon_f$-far distributions and the sample complexity depends on the difference $\rho=\varepsilon_f-\varepsilon_c$.

Specifically, the protocol uses $\tilde{O}(\sqrt{N}/\rho^2)$ samples from $D$, which matches that of prior work by [Chiesa and Gur 2018] and is quadratically better than lower bounds for standalone (i.e without a prover) tolerant distribution testing problems. The main novelty of this work lies in the quadratic improvement of the protocol’s communication complexity and running time over prior work. This is achieved using cryptographic collision resistant hash functions. The prover is assumed to have knowledge of the distribution $D$ up to sufficient accuracy and forced to commit to a distribution as part of the protocol without having to transmit an explicit description of it. This fact along with the use of the PCP theorem are the tools that make it possible for the communication complexity and running time to be significantly reduced. The protocol works by first having the prover send a short digest $d$ of the distribution $D$ using cryptographic hashing to commit to it for the answers to future queries. Subsequently, the verifier uses queries to run an interactive argument of proximity (IAP) in order to check if the bit string representation of the committed distribution is accepted by a turing machine that decides closeness to the property.

The above protocol provides a unified way of dealing with any class/property of distributions that is decidable in polynomial time and since there is no sampling involved in that step, the sample complexity is the same for every such property.

**Strengths:**

The paper cleverly combines tools from cryptography with interactive proof protocols to achieve sublinear communication complexity and running time for a fundamental class of problems.

**Weaknesses:**

In my opinion, the results, techniques used in the paper as well as the lack of experimental evaluation, would make it a better fit for a theoretical computer science conference.


Minor comments:


-Line 20: “should possible”->“should be possible”
-Line 125-128: This does not seem to give a lower bound for the distance $\delta$. You need to run tolerant verification with $\varepsilon_c=\delta-\rho$ and $\varepsilon_f=\delta$ as well.
-Line  184: It seems that the dependence on the parameter $\varepsilon$ is missing from the sample complexity that is cited here.

**Questions:**

Could your techniques be applied to property testing where there are no samples involved (e.g graph property testing) by addressing the queries to the prover?
Can you elaborate more on the benefit of this work to the community of ICLR in particular?

---

> ### Author Response · Authors · 2024-11-24
>
> Thank you for your insightful feedback, as well as the minor comments, that we now took into account. Regarding your concern about the paper’s fit for ICLR: we agree that the tools and results are theoretical, but we believe our work puts forward a novel direction for addressing concerns about trustworthiness of data-driven computations: a core issue for the ICLR community. We hope that including our work in the conference program can lead to future work that builds on our theoretical contributions and brings them closer to practicality. This has happened in related areas, where theoretical proof-system protocols are playing a role in real-world systems (e.g. in block-chain settings).
>
> As to your question about other settings, and in particular allowing query access to the input: if the verifier had query access to its input, be it a graph or a boolean function, the problem becomes easier and there is a body of previous work in this model (see Section 3.3) would allow efficient verification. The main challenge in our work is that sample access is much more restrictive.

---

### Official Review · Reviewer_m7Ct · 2024-11-10

**Soundness:** 3
**Presentation:** 2
**Contribution:** 3
**Rating:** 6
**Confidence:** 3

**Summary:**

This paper falls in the area of distribution testing. In the standard setting of distribution testing, we have given sample access to an unknown distribution $D$ over a finite domain of size $N$ and the goal is to design algorithm that tests whether $D$ is $\epsilon_1$-close (in some distance measure such as total variation distance) to a distribution  satisfying a property ${\mathcal P}$ or $D$ is $\epsilon_2$-far from any distribution that satisfies ${\mathcal P}$. It is known that in this (known as the tolerant testing) setting, the sample complexity, even for very simple properties such as testing uniformity, is close to $\theta(N/\epsilon^2)$.

The paper argues that if we take certain interactive proof system approach the sample complexity can be improved by a quadratic factor. In this model, there is a prover and a verifier both of them having sample access to the unknown distribution. An interactive proof is a communication protocol between prover and verifier so that:

1. if the distribution $D$ is close to having the property ${\mathcal P}$, then an efficient prover can convince the verifier of this fact with high probability (completeness).

2. However, If $D$ is far from having the property ${\mathcal P}$, then no polynomial time prover can make the verifier accept with non-negligible probability (soundness).

The main result is that for any ${\mathit efficiently}$ decidable property, there is such a proof system (under standard cryptographic assumptions) with verifiers sample and time complexities $\tilde{O}(\sqrt{N}/\epsilon^2)$. This is a quadratic improvement from the standard scenario. The paper also discusses several extensions.

**Strengths:**

The model and the results are interesting and believe that  this paper will be of interest to those working in the property testing area.

**Weaknesses:**

The paper appears to have been hastily written. Even some of the main definitions are relegated to the appendix. The main body of the paper primarily discusses rough ideas, with only one formal statement of the main result. Furthermore, there are no formal proofs provided (I cannot see them in the appendix also). While the main ideas are described clearly, the paper lacks the level of technical clarity that the reviewer would expect from an ICLR submission.

**Questions:**

I do not have any specific questions regarding the content, and I appreciate the results presented in the paper. However, I strongly recommend that the authors focus on improving the clarity and formalism in the presentation. While the core ideas are interesting, the current exposition may be challenging for readers who are not deeply familiar with this specific line of research.

---

> ### Author Response · Authors · 2024-11-24
>
> Thank you for your insightful feedback. We are gratified that you found the model and results interesting. To address issues with the writing and the formalism, we revised the paper to include more of the formal definitions and statements in the appendices (end of Appendix B, and the entirety of Appendices C and D).

---

> > ### Comment · Reviewer_m7Ct · 2024-11-27
> >
> > Thanks for making major updates to the paper to include more proofs. I have updated the score.

---

### Author Response · Authors · 2024-11-25

We would like to thank the reviewers for their thoughtful and insightful feedback. Following multiple comments, we expanded on the existing appendices, and added two additional ones (Appendices C and D) to capture more technical details of the protocol and the proofs behind it. We hope that these additions will help communicate our result better.

---

### Meta-Review · Area_Chair_5KLc · 2024-12-17

**Metareview:**

This paper considers the problem of verifying whether an unknown discrete distribution satisfies certain properties with a prover. The authors show that it is possible to verify a wide class of properties using a sublinear number of samples and running time. Without a prover, even testing some properties requires quasi-linear samples, which is a huge improvement.

All reviewers converged in praise of the significance of the result and gave positive scores. I thus recommend acceptance and a spotlight presentation.

**Additional Comments On Reviewer Discussion:**

The authors enhanced the clarity of the exposition in the paper, which was appreciated by reviewers.

---

### Decision · Program_Chairs · 2025-01-22

Accept (Poster)